# SEMANTIC CALIBRATION IN MEDIA STREAMS

## ABSTRACT

Current generative models can produce synthetic media that is visually indistinguishable from real content. As a result, traditional detection methods rely mostly on subtle artifacts introduced during generation. However, we show that such methods could eventually become ineffective. Anticipating this, we suggest that the main risk lies not in whether a media sample is synthetic or real, but in whether its semantic content is *deceiving*, that is, whether it distorts the information distribution in a way that misrepresents reality. To capture this, we formally introduce the notion of *deception* in the context of online media streams. Complementing standard detection approaches, we introduce *semantic calibration* to mitigate deception directly by processing semantic content using captioning and large language models, rather than relying on artifacts introduced by generative models. Our method is explainable, transparent, and modality agnostic, providing a rigorous foundation for developing new tools to combat online misinformation. We offer both theoretical justification and empirical evidence for its effectiveness.

## 1 INTRODUCTION

Artificial intelligence has opened a new front in the war against misinformation on online platforms (Day, 2019). Modern generative tools are now capable of producing synthetic media, sometimes called *deepfakes* (Ramesh et al., 2022; Rombach et al., 2022; Saharia et al., 2022), that is indistinguishable from real content to the human eye. As a result, detecting these digital forgeries depends increasingly on heuristics and learned methods (Mirsky & Lee, 2021; Verdoliva, 2020; Rana et al., 2022; Heidari et al., 2024). While it may initially appear that the arms race between generation and detection is balanced (i.e., that advances in generative capabilities can, in principle, be met with corresponding progress in detection techniques), we show in this work that under mild assumptions, any detection methods that rely solely on invisible artifacts (i.e., non-semantic content) could eventually become obsolete, echoing concerns that current detection cues may not remain reliable (Wang et al., 2020; Corvi et al., 2023).

While this may seem like a discouraging conclusion, we argue that the ultimate goal is not to simply determine whether a media sample is real or synthetic, but rather to assert if its semantic information can be trusted. Detecting synthetic media serves only as a *proxy* for identifying (and potentially removing) deceptive information, based on the assumption that real media reliably reflects true events, but synthetic media does not necessarily convey false information. For example, refining a real but low-resolution image with superresolution tools produces a synthetic image without altering the original semantics (Ledig et al., 2017; Saharia et al., 2023). Conversely, real content can be overrepresented or taken out of context (so-called *cheapfake* (Paris & Donovan, 2019)), posing risks similar to those of synthetic media. Additionally, the boundary between real and generated media is often blurred. For instance, studies have shown that diffusion models can memorize and generate images from their training sets (Somepalli et al., 2023a;b). If a generated image is a pixel-perfect replica of a real photo, should it still be considered fake?

Motivated by these observations, we present in this work an alternative, holistic perspective on the issue, by shifting the focus from the *provenance* of a media sample (was it synthetically generated?) to its *semantic plausibility* (is it surprising beyond credible bounds?). We refer to this property as *deception*, capturing the idea that deepfakes and cheapfakes can distort the distribution of semantic information across media streams, and therefore deceive. Building on this perspective, we introduce *semantic calibration*, a method designed to mitigate this risk, even in a future where generative models are no longer detectable. Our approach first converts the semantic content of a media sample

This image shows a `male` `tennis` player in `action` on a `clay` court . He is wearing a blue shirt , `white` shorts , and `white` sneakers . `The` player is holding a `red` and black `tennis` `racket` and is in the `middle` of a `forehand` swing `,` with his body `stretched` out to `hit` the ball . His `shadow` can be seen on the `ground` , `indicating` that he is in `motion` . The `court` `appears` to be `well` - maintained and the `clay` is a reddish - brown color .

This image shows two `young` men `sitting` on a `couch` in a living room . They are both `wearing` `white` `shirts` and black `ties` . The man on the left is `wearing` `glasses` and is smiling at the camera . `He` is holding a white `mug` in his `hand` and appears to be `giving` `it` to the other man . `There` is a `basket` of `flowers` on the `couch` next to them . `In` the background , there is a `piano` and a books he lf with various `items` on it .

Figure 1: Saliency map showcasing the explainability of semantic calibration on two test images from the COCO dataset. We simulate a setting where sport-related activities are considered fake, i.e. $q(z = \text{sport}) > p_r(z = \text{sport}) = 0$, by creating a dataset $\mathcal{D}_r$ with no sport-related labels. Tokens highlighted in **blue favor acceptance** ($\Delta_i > 0$) while those highlighted in **orange favor rejection** ($\Delta_i < 0$). For clarity, tokens belonging to the same word are merged. As expected, the algorithm rejects the first image and accepts the second with probability $\approx 1$. More importantly, it bases its decisions on words that are intuitive for the given setting (e.g., `tennis`, `forehand`, `stretched` for rejection, and `sitting`, `couch`, `glasses` for acceptance). For the same two images, we also plot the rolling acceptance probabilities in Fig. 5 for three milder semantic shifts $q(z = \text{sport}) > p_r(z = \text{sport}) > 0$.

into text using a captioning model, then applies rejection sampling in this textual space. Acceptance probabilities are computed using two fine-tuned LLMs trained to model the semantic distributions.

Although the distinction between semantic and non-semantic content may seem arbitrary, it reflects how modern generative models are trained (Ramesh et al., 2022; Rombach et al., 2022; Saharia et al., 2022). Semantic content can be understood as the conditioning signal (i.e., prompts), while non-semantic content includes the low-level details the model autonomously fills in to generate realistic outputs. In high-dimensional modalities such as images or video, non-semantic components such as grass patterns, cloud shapes, or water textures dominate the overall data. We estimate in Appendix A.1 that for a small natural image, only about 3% of the information perceived by a human is semantic. Due to this imbalance, traditional deepfake detection methods have remained largely content-agnostic, relying instead on non-semantic artifacts for detection (Frank et al., 2020; Durall et al., 2020; Rössler et al., 2019). However, such approaches fundamentally rely on the imperfections of current generative models, which future models may no longer exhibit.

For these reasons, it is important to explore alternative strategies. We propose reframing deepfakes not as a binary issue of authenticity, but as a distributional problem of semantic information, where the primary risk lies in how a media sample shifts this distribution, regardless of whether it is real or synthetic. To the best of our knowledge, this is the first work to formalize deception in distributional terms and to present an explainable approach for mitigating it. We summarize our contributions as follows:

- First, under mild assumptions and drawing on classical results from hypothesis testing, **we show that traditional deepfake detection methods could eventually become obsolete** if they rely purely on non-semantic artifacts.

- Anticipating this scenario, **we adopt a holistic, distributional perspective and introduce the concept of *deception*:** the strategic manipulation of the semantic distribution of media streams. Our analysis shows that deepfake detection serves only as a proxy for our primary goal of reducing semantic deception.

- Finally, **we introduce a method that targets deception directly** by filtering media based on their semantic content. Our approach converts media into text via a captioning model, then applies rejection sampling using two fine-tuned LLMs modeling the semantic distributions. We further provide theoretical evidence that our formulation is well-posed, validate our methods empirically, and explore the explainability of our approach.

## 2 BACKGROUND

**Media stream.**   We consider a setting where various forms of media (e.g., images, videos, audio) are uploaded to an online platform and redistributed to users of that platform. The core of our analysis is built on a set of random variables that model the media stream. Specifically, $X \in \mathcal{X}$ denotes the media itself. Each media item carries semantic content represented by a latent variable $Z \in \mathcal{Z}$ that is deterministically determined by $X$, i.e., $Z = f(X)$. To simplify our analysis, we consider that both $\mathcal{X}$ and $\mathcal{Z}$ are discrete sets (e.g., digital content). We introduce a binary variable $G \in \{0, 1\}$ indicating whether the media is real ($G = 0$) or generated ($G = 1$). From these random variables, we define the *media stream* $p$ as the joint distribution

$$p(g, z, x) = \mathbb{P}\left[G = g, Z = z, X = x\right].$$

When the context is clear, we also use $p$ to refer to the marginal distribution over any subset of these variables (e.g., $p(x) = \mathbb{P}\left[X = x\right]$), and we denote by $p_G = \mathbb{P}\left[G = 1\right]$ the prior probability of observing a generated sample in the media stream $p$. We assume $0 < p_G < 1$. Additionally, we denote by $q(x, z) = \mathbb{P}[X = x, Z = z \mid G = 1]$ the distribution of generated media samples and by $p_r(x, z) = \mathbb{P}[X = x, Z = z \mid G = 0]$ the distribution of real media samples. With this notation, the overall media stream distribution $p$ can be written as the mixture

$$p(z, x) = p_G q(z, x) + (1 - p_G) p_r(z, x). \tag{1}$$

See Appendix A.2 for a detailed summary of the notation used in this work.

**Generative models.**   Let $q(x|z) = \mathbb{P}[X = x \mid Z = z, G = 1]$ denote the conditional generative model used to generate synthetic media. This may correspond to a neural generative model or an ensemble of such models. These generators are typically trained to minimize the following quantity (or a relaxed proxy):

$$\mathcal{L}(q, p_r) \triangleq \sup_{z \in \mathcal{Z}} D_{\mathrm{KL}}(q(x|z) \| p_r(x|z)), \tag{2}$$

where $p_r(x|z)$ denotes the distribution of real media that conveys semantic information $z$. To ensure that $p_r(x|z)$ and $q(x|z)$ are well-defined, we assume that $p_r(z), q(z) > 0$ for any $z$. See Appendix A.3 for a discussion about the alternative (forward) objective $\mathcal{L}(p_r, q)$.

**Deceptive vs neutral media stream.**   We say that a media stream $p$ is *deceptive* if $p(z) \neq p_r(z)$. This directly implies the following factorization (illustrated in Fig. 2):

$$p(z, x) = p_G q(x|z) q(z) + (1 - p_G) p_r(x|z) p_r(z).$$

Conversely, if no dependency is assumed between the generation label $G$ and the semantic content $Z$, such that there is no edge between $G$ and $Z$ in Fig. 2, we refer to the resulting media stream as *neutral*. To quantify the deviation from neutrality, we define the *deception* of a media stream $p$ as

$$\delta(p \mid p_r) \triangleq D_{\mathrm{KL}}(p(z) \| p_r(z)). \tag{3}$$

**Reducing deception.**   One objective of content moderation is to reduce the deception of a media stream $p$ using a decision rule $\phi(x) = \mathbb{P}[F_\phi = 1 \mid X = x]$, where $F_\phi = 1$ means that the content is flagged as deceptive. Moderation then happens by removing flagged content, which results in the filtered media stream

$$p^\phi(g, z, x) = \mathbb{P}\left[G = g, Z = z, X = x \mid F_\phi = 0\right], \tag{4}$$

Using the convexity of the KL divergence (Cover & Thomas, 2006), we show in Appendix A.4.1 that

$$\underbrace{\delta(p^\phi \mid p_r)}_{\text{direct objective (ours)}} \quad \leq \quad \underbrace{p_G^\phi \delta(q^\phi \mid p_r) + (1 - p_G^\phi) \delta(p_r^\phi \mid p_r)}_{\text{proxy objective (deepfake detection)}}, \tag{5}$$

where the superscript $\phi$ indicates conditioning on $F_\phi = 0$ (e.g., $p_G^\phi = \mathbb{P}[G = 1 \mid F_\phi = 0]$, see Appendix A.2 for more details on the notation). From Eq. (5), one can identify two avenues to decrease the deception of $p^\phi$. The first is to build a decision rule that minimizes the right-hand side of the inequality by reducing $p_G^\phi$ while keeping $p_r^\phi(z)$ close to $p_r(z)$ such that $\delta(p_r^\phi \mid p_r) \approx 0$. This is precisely the aim of traditional deepfake detectors that rely on non-semantic cues. However, as we show in the next section, this paradigm could become ineffective if generative models continue to improve. In addition, minimizing the right-hand side of Eq. (5) is only a proxy to the primary objective, which is to minimize $\delta(p^\phi \mid p_r)$ directly. Therefore, we propose exploring a more direct alternative in this paper, which is to bring $p^\phi(z)$ closer to $p_r(z)$ without explicitly targeting $p_G^\phi$.

**Limits of deepfake detection.** We frame deepfake detection as a binary hypothesis testing problem where the classification error is captured by the indicator variable $E_\phi = \mathbb{1}\{F_\phi \neq G\}$. Given a media stream $p$, the objective of deepfake detection is to find a decision rule $\phi$ that maximizes the expected accuracy $\mathbb{P}[E_\phi = 0]$. See Fig. 2 for a detailed view of how deepfake detection integrates with the media stream $p$. As conditional generative models continue to improve their ability to replicate reality, it is reasonable to assume that, for any $\epsilon > 0$, a generator $q$ will eventually exist such that $\mathcal{L}(q, p_r) \leq \epsilon$. Based on this assumption, we derive a fundamental lower bound on the maximal achievable accuracy of any decision rule.

**Theorem 1.** *Let $q(z, x) = q(x \mid z)q(z)$ be such that $\mathcal{L}(q, p_r) \leq \epsilon$. Then*

$$\sup_\phi \mathbb{P}\left[E_\phi = 0\right] \leq \max\{p_G, 1 - p_G\} \left(1 + \sqrt{\frac{\epsilon + \delta(q \mid p_r)}{2}}\right). \tag{6}$$

See Appendix A.4.3 for the full proof. This bound highlights that as generative models improve (i.e., $\epsilon \to 0$), the accuracy of any detector becomes primarily limited by the deception of the media stream. Consequently, it will become increasingly difficult to construct semantic-agnostic decision rules $\phi$ that perform well across arbitrary semantic distributions $q(z)$ (i.e., such that $p_G^\phi \ll p_G$).

## 3 METHOD

**Calibration.** Our goal is to design a decision rule $\phi$ that filters a deceptive media stream $p$ into a neutral stream $p^\phi$, as per Eq. (4), such that $\delta(p^\phi \mid p_r) \approx 0$. We refer to this process as *semantic calibration*. The main challenge is that we do not sample directly from $p_r$, but from $p$. However, for samples $x \sim p$, we assume access to estimates of both $p(z)$ and $p_r(z)$, where $z = f(x)$ denotes the semantic content of $x$ (see below for details on how $f(x)$, $p(z)$, and $p_r(z)$ are approximated in practice). When a media $x$ with semantics $z = f(x)$ is sampled from the full stream $p$, we accept it with probability $p_r(z)/Mp(z)$, where $M \geq M^* \triangleq \sup_{z' \in \mathcal{Z}} p_r(z')/p(z')$. Using this rejection sampling procedure, it can be shown that $p^\phi(z) = p_r(z)$ (see Appendix A.4.4).

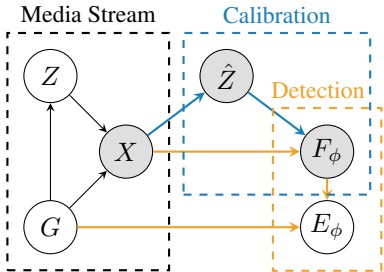

Media Stream    Calibration

Detection

Figure 2: Bayesian network showing the factorization of both deepfake detection and semantic calibration. Observed variables are shown in gray.

**Computing $M^*$.** The constant $M^*$ is well-defined only if the support of $p(z)$ is contained within the support of $p_r(z)$. Moreover, computing $M^*$ directly appears intractable, as it would require evaluating the ratio $p_r(z)/p(z)$ for every possible semantic representation $z$. Fortunately, in our setting, $p(z)$ is a mixture of $q(z)$ and $p_r(z)$ as per Eq. (1). This implies that for any $z$ such that $p_r(z) > 0$, we also have $p(z) > 0$ (assuming $p_G < 1$). In fact, it can be shown that $M^* \leq 1/(1 - p_G)$ (see Appendix A.4.5). Hence, choosing $M \geq 1/(1 - p_G)$ ensures that $p^\phi(z) = p_r(z)$. If $p_G$ is unknown, a conservative estimate $\hat{p}_G \geq p_G$ can be used instead.

**Estimating $p_r(z)/p(z)$.** While $X$ is observed, $Z$ is latent and it remains unclear how to compute the ratio $p_r(z)/p(z)$ for a given $(x, z) \sim p$. Since $z$ is the semantical information of a media, we propose to approximate it using a textual representation (i.e. a finite series of $n$ tokens $\hat{Z} = \hat{Z}_1 \hat{Z}_2 \cdots \hat{Z}_n$). To that end, we use a captioning model $\hat{f}$ to estimate $f$. Ideally, $\hat{z} = \hat{f}(x)$ captures all semantic information a human would perceive in media $x$ (formally, we aim for $H(Z \mid \hat{Z}) = 0$, where $H(Z \mid \hat{Z})$ denotes the entropy of $Z$ given $\hat{Z}$). To achieve this, the model should be biased toward generating detailed captions. This reduces the risk of omitting semantically relevant details. However, because the mapping $z \mapsto \hat{z}$ is one-to-many, the ratios $p_r(\hat{z})/p(\hat{z})$ might differ from $p_r(z)/p(z)$. Fortunately, under the assumption of perfect generation, i.e., $q(x|z) = p_r(x|z)$ for any $z$, we show that both ratios are equal (see Appendix A.4.6).

**Estimating $p_r(\hat{z})/p(\hat{z})$.** Once $\hat{z} = \hat{f}(x)$ is computed, we estimate $p_r(\hat{z})$ and $p(\hat{z})$ using language models $\pi_{\theta_r}(\hat{z})$ and $\pi_\theta(\hat{z})$ that were auto-regressively trained on the captioned data from the real and full media streams ($\mathcal{D}_r$, respectively $\mathcal{D}$). See Section 6 for more details on how $\mathcal{D}$ and $\mathcal{D}_r$ can be

constructed in practice. Finally, we perform rejection sampling in the space of textual approximations $\hat{z}$, using the ratio $\pi_{\theta_r}(\hat{z})/\pi_\theta(\hat{z})$ to approximate $p_r(z)/p(z)$. For numerical stability, we compute this ratio in log-space and only consider a subset of tokens $\mathcal{I}_\rho(\hat{z}) \subseteq \{1, \cdots, |\hat{z}|\}$ determined by a parameter $\rho$ (see Section 4), leading to the following definition:

$$r(\hat{z}; \pi_\theta, \pi_{\theta_r}, \rho) \triangleq \exp\left(\sum_{i \in \mathcal{I}_\rho(\hat{z})} \left[\log \pi_{\theta_r}(\hat{z}_i \mid \hat{z}_{<i}) - \log \pi_\theta(\hat{z}_i \mid \hat{z}_{<i})\right]\right) \approx \frac{p_r(\hat{z})}{p(\hat{z})}. \tag{7}$$

**Proposed method.** Given any media sample $x$, a captioning model $\hat{f}$, two language models $\pi_\theta$ and $\pi_{\theta_r}$, and a conservative estimate $\hat{p}_G \geq p_G$, we define the decision rule

$$\phi(x; \pi_\theta, \pi_{\theta_r}, \hat{f}, \hat{p}_G, \rho) \triangleq \min\left\{(1 - \hat{p}_G) \cdot r(\hat{f}(x); \pi_\theta, \pi_{\theta_r}, \rho), 1\right\}, \tag{8}$$

where the clipping ensures the output remains a valid probability by accounting for numerical and approximation errors. Then, when a media sample $x$ is uploaded to the media stream, we sample $F_\phi \sim \text{Bernoulli}(\phi(x; \pi_\theta, \pi_{\theta_r}, \hat{f}, \hat{p}_G))$, and filter out $x$ when $F_\phi = 1$.

# 4 EXPERIMENTS

**Overview.** We conduct a series of experiments to evaluate the effectiveness of our decision rule for mitigating deception in media streams. To simulate the case where generation is perfect, we rely exclusively on real text, image, and audio datasets, therefore ensuring $\mathcal{L}(q, p_r) = 0$. Firstly, we quantitatively evaluate our semantic calibration method. This allows us to work with known semantic distributions and precisely control distributional shifts. We create these synthetic shifts using labeled datasets and compute deception based on changes in label distributions. Secondly, since real-world

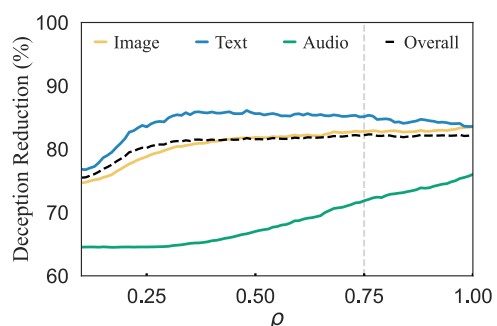

Figure 3: Filtering performance across $\rho$ values. We use $\rho = 0.75$ for text and image, $\rho = 1$ for audio.

scenarios typically lack access to such ground-truth labels, we demonstrate that our method offers high explainability and transparency, making it practical even when deception cannot be directly measured.

**Tractable semantic distributions.** To create a setting where deception can be evaluated, we use the labeled categorical datasets listed below and assume that the latent semantic variable $Z$ corresponds to the class label. The semantic distribution $p(z)$ is therefore the distribution over these labels, which allows us to create a setting where deception is tractable. We construct two versions of each dataset: one for training the *general* model $\pi_\theta(\hat{z})$, denoted by $\mathcal{D}$, and one for training the *real* model $\pi_{\theta_r}(\hat{z})$, denoted by $\mathcal{D}_r$. Semantic distortion is introduced by reweighting the label frequencies in $\mathcal{D}_r$. For each dataset, we set target class proportions and fill the real training set using as much data as possible without exceeding the specified counts. We generate multiple distorted versions per dataset, each corresponding to a different label distribution. The severity of each shift is quantified using $\delta(p \mid p_r)$.

**Datasets.** Our experiments span four modalities: text, image, video, and audio. For text, we use Senator Tweets (KristinCoyote, 2025) with conservative/liberal labels, Political Podcasts (nbandhi, 2025) with again the same labeling. We also use Political Bias (surajkarakulath, 2025) with labels left/center/right, AG-News (Zhang et al., 2015), labeled across four categories, and a custom dataset from eight distinct books from Project Gutenberg (Project Gutenberg, 2025), where semantically split phrases are labeled by book origin. For images, we use CIFAR-10 (Krizhevsky, 2009) (10 classes), CIFAR-100 (Krizhevsky, 2009) (100 classes), a coarse-labeled ImageNet (Deng et al., 2009) subset (20 custom coarse classes), and a COCO (Lin et al., 2014a) subset with a binary coarse labeling. We also apply our framework in the multi-modal setting of images with text. For this, we use Hateful

Memes Challenge Kiela et al. (2021) with existing labels (hateful and non-hateful). For video, we use ActivityNet (Caba Heilbron et al., 2015), with 52 coarse labels. For audio, we use VGGSound (Chen et al., 2020) (14 custom coarse classes) and UrbanSound8k (Salamon et al., 2014) (10 original classes). A detailed description of each dataset and labeling scheme for each experiment can be found in Appendix A.5. We generate image captions using the Florence (Yuan et al., 2021) model for all datasets. For audio, we use Qwen-Audio (Chu et al., 2023) to caption UrbanSound8k and reuse AudioSetCaps (Mei et al., 2023) captions for VGGSound. The exact prompts used to caption the different modalities can be found in Appendix A.6. Each original dataset is first split 80/20 into general training and test subsets, and the general training set is then further split 80/20 into training ($\mathcal{D}$ and $\mathcal{D}_r$) and validation subsets.

**Training.** Our pipeline begins by translating media into text where applicable (e.g., images via the Florence captioning model (Yuan et al., 2021), audio via Qwen-Audio (Chu et al., 2023)). We then fine-tune two GPT-2 Small models (Radford et al., 2019) (124M parameters), one on $\mathcal{D}$ and one on $\mathcal{D}_r$, both initialized from pre-trained weights. Smaller LLMs are easier to train for our objective, especially given the limited training data. Both models are trained for 5 epochs using AdamW (Loshchilov & Hutter, 2019) with a learning rate of $2\mathrm{e}{-5}$, a linear scheduler, and 1000 warmup steps. The checkpoint with the best validation loss is selected. Full implementation details can be found in Appendix A.7.

**Inference.** During inference, we evaluate the likelihood ratio of each text or caption $\hat{z} = \hat{f}(x)$ from the general test set using both models $\pi_\theta(\hat{z})$ and $\pi_{\theta_r}(\hat{z})$, following Eq. (7). To reduce noise from low-impact tokens and focus on the most informative parts of the sequence, we apply a top-$\rho$ filtering strategy over the log-probability differences, inspired by nucleus sampling (Holtzman et al., 2020). Formally, let

$$\Delta_i \triangleq \log \pi_{\theta_r}(\hat{z}_i \mid \hat{z}_{<i}) - \log \pi_\theta(\hat{z}_i \mid \hat{z}_{<i}), \tag{9}$$

and let $\omega$ be a permutation of $\{1, 2, \cdots, |\hat{z}|\}$ such that $|\Delta_{\omega(1)}| \geq |\Delta_{\omega(2)}| \geq \cdots \geq |\Delta_{\omega(|\hat{z}|)}|$. We define $k_\rho$ as the smallest $k$ such that $\sum_{j=1}^{k}|\Delta_{\omega(j)}| \geq \rho \cdot \sum_{i=1}^{|\hat{z}|}|\Delta_i|$, and we use $\mathcal{I}_\rho(\hat{z}) \triangleq \{\omega(j) \mid 1 \leq j \leq k_\rho\}$ in Eq Eq. (7). This limits the computation to tokens that contribute most to the distributional shift, improving robustness in long captions. See Fig. 3 for an analysis of the optimal $\rho$ value for each modality. Finally, we simply use our decision rule in Eq. (8) to compute the probability of flagging the media sample.

## 5 RESULTS

**Quantitative.** We present the main results of our experiments in Table 1. For each dataset, we evaluate performance across four levels of semantic shift (defined as the deception of the unfiltered general distribution $\delta(p \mid p_r)$), including a baseline serving as a no-shift control. For the baselines, we observe that calibration introduces little to no deception ($\leq 0.103$, which, for comparison, is less than half the level observed in the most deceptive mild-shift setting), confirming that the method does not introduce unwanted semantic bias when none exists. As the semantic shift increases (from mild to severe), our method consistently reduces deception across all datasets and modalities, with reductions ranging from 64% to as high as 99%. Calibration is especially effective for long-text modalities (text and images) and remains robust for more compact representations like audio captions.

**Qualitative.** To complement scalar metrics and provide an intuitive understanding of what different levels of deception represent, we visually compare in Fig. 4 the filtered and real distributions $p^\phi$ and $p$, respectively. These plots show that our method reliably aligns the filtered distributions with the intended targets, across all modalities and even under strong semantic shifts. We emphasize that for all results presented in Table 1 and Fig. 4, labels are only used to measure deception reduction. They are never used during training, reflecting the fact that $Z$ is latent, as illustrated in Fig. 2. This demonstrates the potential of our approach to filter media streams based on intractable latent semantic distributions.

**Explainability.** The qualitative and quantitative results presented above are made possible by a simplified, controlled setting where the semantic space $\mathcal{Z}$ is small, discrete, and where the distributions $p_r(z)$ and $p(z)$ are known. In real-world applications, these semantic distributions are

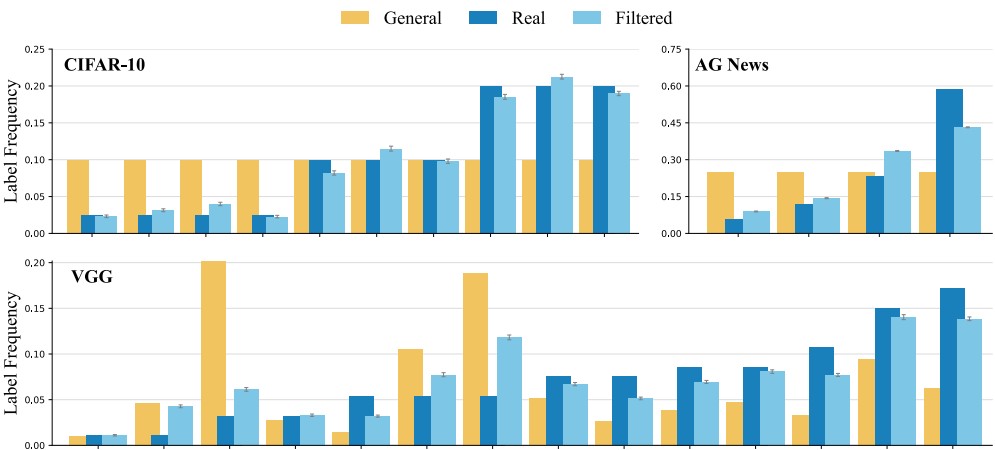

Figure 4: Visualization of the **filtered distribution** $p^\phi(z)$ compared to the **real distribution** $p_r(z)$ and the **general distribution** $p(z)$. The plots show the moderate shift scenario for three datasets CIFAR-10, AG-News, and VGG-Sound, each representing a different modality. The positions along the horizontal axis correspond to class labels, ordered by increasing frequency under $p_r(z)$. We observe that $p^\phi(z)$ generally lies between $p(z)$ and $p_r(z)$, but most closely resembles the latter.

intractable, making direct quantitative evaluation of deception impossible. Nonetheless, our approach provides high explainability and transparency, offering actionable insights even when exact semantic distributions cannot be computed. This constitutes a strong advantage compared to more opaque media stream moderation algorithms. More precisely, we can gain fine-grained insight into the impact of each word by analyzing the per-token log-probability differences $\Delta_i$. This makes it possible to identify which parts of the caption are semantically more typical of the real distribution versus the general one, shedding light on how our filtering algorithm makes its rejection decisions. For instance, Fig. 1 presents a token-level saliency map for two COCO images, highlighting the semantic cues the algorithm relies on to make its decisions. Additional figures can be found in the Appendix A.11. Additionally, Fig. 5 illustrates how the severity of the semantic shift affects rejection behavior, with milder shifts leading to more uncertain acceptance or rejection decisions, as expected.

## 6 DISCUSSION AND LIMITATIONS

**Relation to out-of-distribution (OOD) detection.** OOD detection is the binary task of determining whether a sample comes from a reference distribution. When operating in the semantic space $\mathcal{Z}$, OOD methods typically rely on $p_r(z)$, classifying a media sample $x$ with semantics $z = f(x)$ as OOD whenever $p_r(z) < \tau$ for a given threshold $\tau > 0$ (methods that do not use $p_r$ are quite disconnected from our setting). If $x$ is observed twice, it is expected that the detection mechanism will come up with the same conclusion. Each sample $x$ is treated independently and the actual distribution of $x$ is typically of no concern. Semantic calibration, on the other hand, is interested in the ratio $p_r(z)/p(z)$, with $p(z)$ being a reference distribution. If both $p_r(z)$ and $p(z)$ are small but similar, then calibration might accept the media $x$, whereas an OOD detector might reject the sample due to its low semantic probability. However, there is a special case in which semantic calibration coincides with OOD detection, namely when $p_r$ and $q$ have disjoint support. In that case, $p_z = (1 - p_G)p_r(z)$ for samples

Table 2: Detection performance of Eq. (8) on misinformation and hate detection datasets: ISOT (Ahmed et al., 2017), LIAR (Wang, 2017), FAKE TFG (Álvarez Hervás, 2022) and Hateful Memes Kiela et al. (2021).

| Dataset | Acc. | Recall | Prec. | F1 |
|---|---|---|---|---|
| ISOT | 0.998 | 0.999 | 0.997 | 0.998 |
| LIAR | 0.869 | 0.851 | 0.952 | 0.887 |
| Fake TFG | 0.972 | 0.977 | 0.972 | 0.974 |
| Hateful Memes | 0.842 | 0.890 | 0.729 | 0.802 |

Table 1: Semantic calibration performance across various datasets and semantic shifts, measured by the deception reduction $(\delta(p|p_r) - \delta(p^\phi|p_r))/\delta(p|p_r)$. Errors in italics indicate *standard deviation over 1000 inference runs* (for a single training run), while errors in underline indicate standard deviation over 3 training runs (for a single inference run). We report training errors for severe shifts only to save computation, as similar errors were observed across other shifts. We use the conservative estimate $\hat{p}_G = 1 - |\mathcal{D}_r|/|\mathcal{D}|$. The shifts shown in **blue** can be visualized in Fig. 4.

| | Dataset | $\|\mathcal{D}_r\|$ | $\hat{p}_G$ | Shift | $\delta(p \mid p_r)$ | $\delta(p^\phi \mid p_r)$ | Reduction |
|---|---|---|---|---|---|---|---|
| **Text** ($\rho=0.75$) | **Senator Tweets** | 31.6k | 0.60 | Baseline | 0.000 | 0.000 ± *0.000* | - |
| | $\|\mathcal{Z}\|=2$ | 35.6k | 0.55 | Mild | 0.020 | 0.001 ± *0.002* | 96.03% |
| | $\|\mathcal{D}\|=79.2k$ | 33.3k | 0.58 | Moderate | 0.193 | 0.016 ± *0.001* | 91.55% |
| | $\mathbb{E}[\|\hat{Z}\|]=41.3$ | 23.8k | 0.70 | Severe | 0.368 | 0.067 ± *0.002* / 0.004 | 81.99% |
| | **Political Podcasts** | 4.6k | 0.60 | Baseline | 0.000 | 0.001 ± *0.000* | - |
| | $\|\mathcal{Z}\|=2$ | 9.1k | 0.20 | Mild | 0.022 | 0.000 ± *0.000* | 99.77% |
| | $\|\mathcal{D}\|=11.39k$ | 7.1k | 0.38 | Moderate | 0.081 | 0.000 ± *0.000* | 99.99% |
| | $\mathbb{E}[\|\hat{Z}\|]=161.0$ | 5.7k | 0.50 | Severe | 0.207 | 0.013 ± *0.001* / 0.004 | 93.79% |
| | **Political Bias** | 3.8k | 0.60 | Baseline | 0.000 | 0.000 ± *0.000* | - |
| | $\|\mathcal{Z}\|=3$ | 5.7k | 0.41 | Mild | 0.134 | 0.020 ± *0.003* | 84.85% |
| | $\|\mathcal{D}\|=9.59k$ | 3.4k | 0.65 | Moderate | 0.297 | 0.001 ± *0.000* | 97.12% |
| | $\mathbb{E}[\|\hat{Z}\|]=1376.9$ | 3.0k | 0.69 | Severe | 0.368 | 0.057 ± *0.004* / 0.007 | 84.52% |
| | **AG-News** | 38.4k | 0.60 | Baseline | 0.000 | 0.001 ± *0.000* | - |
| | $\|\mathcal{Z}\|=4$ | 48.0k | 0.50 | Mild | 0.239 | 0.014 ± *0.002* | 94.14% |
| | $\|\mathcal{D}\|=96.0k$ | 40.8k | 0.58 | **Moderate** | 0.351 | 0.051 ± *0.001* | 85.47% |
| | $\mathbb{E}[\|\hat{Z}\|]=52.1$ | 34.3k | 0.64 | Severe | 0.603 | 0.006 ± *0.002* / 0.005 | 99.00% |
| | **Gutenberg** | 1640 | 0.60 | Baseline | 0.000 | 0.103 ± *0.009* | - |
| | $\|\mathcal{Z}\|=8$ | 2576 | 0.27 | Mild | 0.254 | 0.052 ± *0.005* | 79.52% |
| | $\|\mathcal{D}\|=4096$ | 1728 | 0.58 | Moderate | 0.563 | 0.071 ± *0.010* | 87.39% |
| | $\mathbb{E}[\|\hat{Z}\|]=109.4$ | 1288 | 0.69 | Severe | 0.860 | 0.014 ± *0.003* / 0.008 | 98.37% |
| **Image** ($\rho=0.75$) | **COCO** | 17.6k | 0.60 | Baseline | 0.000 | 0.000 ± *0.000* | - |
| | $\|\mathcal{Z}\|=2$ | 11.5k | 0.74 | Mild | 0.678 | 0.141 ± *0.007* | 79.20% |
| | $\|\mathcal{D}\|=44.0k$ | 9.5k | 0.78 | Moderate | 1.379 | 0.042 ± *0.004* | 96.95% |
| | $\mathbb{E}[\|\hat{Z}\|]=100.3$ | 9.0k | 0.80 | Severe | 1.926 | 0.009 ± *0.002* / 0.003 | 99.53% |
| | **CIFAR-10** | 16.0k | 0.60 | Baseline | 0.000 | 0.015 ± *0.003* | - |
| | $\|\mathcal{Z}\|=10$ | 13.4k | 0.67 | Mild | 0.237 | 0.009 ± *0.002* | 96.20% |
| | $\|\mathcal{D}\|=40.0k$ | 20.0k | 0.50 | Moderate | 0.347 | 0.009 ± *0.002* | 96.95% |
| | $\mathbb{E}[\|\hat{Z}\|]=93.0$ | 11.5k | 0.71 | Severe | 0.781 | 0.059 ± *0.006* / 0.001 | 92.45% |
| | **CIFAR-100** | 16.0k | 0.60 | Baseline | 0.000 | 0.073 ± *0.007* | - |
| | $\|\mathcal{Z}\|=100$ | 14.3k | 0.64 | Mild | 0.216 | 0.076 ± *0.007* | 64.81% |
| | $\|\mathcal{D}\|=40.0k$ | 14.2k | 0.65 | **Moderate** | 0.403 | 0.080 ± *0.007* | 80.34% |
| | $\mathbb{E}[\|\hat{Z}\|]=91.6$ | 15.2k | 0.62 | Severe | 0.818 | 0.144 ± *0.012* / 0.003 | 80.15% |
| | **ImageNet** | 32.5k | 0.60 | Baseline | 0.000 | 0.020 ± *0.003* | - |
| | $\|\mathcal{Z}\|=20$ | 23.2k | 0.71 | Mild | 0.272 | 0.073 ± *0.003* | 73.16% |
| | $\|\mathcal{D}\|=81.3k$ | 21.0k | 0.74 | Moderate | 0.526 | 0.106 ± *0.003* | 79.85% |
| | $\mathbb{E}[\|\hat{Z}\|]=98.4$ | 12.6k | 0.85 | Severe | 0.796 | 0.083 ± *0.006* / 0.003 | 89.87% |
| **Image + Text** ($\rho=1$) | **Hateful Memes** | 2.7k | 0.60 | Baseline | 0.000 | 0.000 ± *0.000* | - |
| | $\|\mathcal{Z}\|=2$ | 3.4k | 0.50 | Mild | 0.280 | 0.001 ± *0.000* | 99.53% |
| | $\|\mathcal{D}\|=6.8k$ | 2.9k | 0.57 | Moderate | 0.580 | 0.015 ± *0.000* | 97.40% |
| | $\mathbb{E}[\|\hat{Z}\|]=109.4$ | 2.8k | 0.59 | Severe | 0.781 | 0.043 ± *0.001* / 0.002 | 94.48% |
| **Video** ($\rho=1$) | **Activity Net** | 4.8k | 0.60 | Baseline | 0.000 | 0.000 ± *0.000* | - |
| | $\|\mathcal{Z}\|=52$ | 9.9k | 0.17 | Mild | 0.295 | 0.024 ± *0.001* | 91.81% |
| | $\|\mathcal{D}\|=11.9k$ | 7.7k | 0.35 | Moderate | 0.599 | 0.037 ± *0.002* | 93.84% |
| | $\mathbb{E}[\|\hat{Z}\|]=9.8$ | 4.7k | 0.61 | Severe | 0.791 | 0.051 ± *0.001* / 0.009 | 93.55% |
| **Audio** ($\rho=1$) | **Urbansound8k** | 1984 | 0.60 | Baseline | 0.000 | 0.011 ± *0.003* | - |
| | $\|\mathcal{Z}\|=10$ | 2296 | 0.54 | Mild | 0.202 | 0.055 ± *0.006* | 72.77% |
| | $\|\mathcal{D}\|=4960$ | 2128 | 0.57 | Moderate | 0.616 | 0.188 ± *0.014* | 69.48% |
| | $\mathbb{E}[\|\hat{Z}\|]=15.1$ | 2160 | 0.56 | Severe | 0.825 | 0.221 ± *0.019* / 0.005 | 73.21% |
| | **VGG-Sound** | 58.3k | 0.60 | Baseline | 0.000 | 0.002 ± *0.000* | - |
| | $\|\mathcal{Z}\|=14$ | 35.5k | 0.76 | Mild | 0.300 | 0.082 ± *0.003* | 72.67% |
| | $\|\mathcal{D}\|=145.6k$ | 32.0k | 0.78 | **Moderate** | 0.639 | 0.091 ± *0.003* | 85.76% |
| | $\mathbb{E}[\|\hat{Z}\|]=13.1$ | 29.1k | 0.80 | Severe | 0.927 | 0.152 ± *0.004* / 0.005 | 83.48% |

in the support of $p_r$, and $p(z) = p_G q(z)$ otherwise (see Eq. (1)). Setting $M = 1/(1 - p_G)$ as mentioned in Section 3, the ratio $p_r(z)/(Mp(z))$ is either 1 (for $z$ in the support of $p_r$) or 0 (for $z$ in the support of $q$). Our filtering rule in Eq. (8) becomes therefore de facto an exact binary, detection

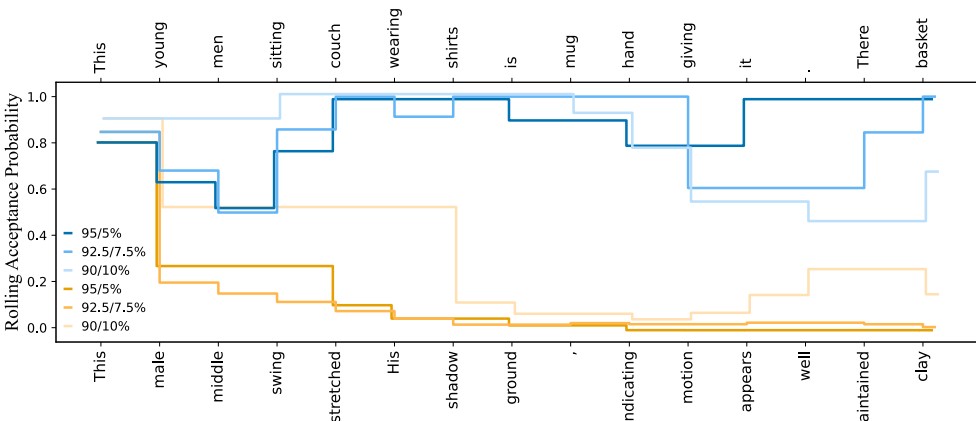

Figure 5: Rolling acceptance probabilities for the two images shown in Fig. 1, in a setting where sport-related content is overrepresented in the general distribution, i.e., $q(z = \text{sport}) > p_r(z = \text{sport})$. Rolling probabilities are computed using Eq. (8) with $\rho = 0.75$, applying a truncated sum in the ratio from Eq. (7). Each image is evaluated under three shift severities in the no-sport/sport scenario. Orange lines show results for the **tennis player**, while blue lines show results for the **two men celebrating**. The general distribution $p(z)$ is 80% no sport, 20% sport.

rule. This special case is precisely the setting of misinformation datasets, as information is either real or fake, but cannot be both (i.e. disjoint support). In Table 2 we present the detection performance of our framework based on our decision rule on a few misinformation and hate detection datasets. The results confirm that our approach generalizes to *semantic detection* tasks on real data.

**Semantic calibration is *not* deepfake detection.** We emphasize that our approach is not designed to perform deepfake detection. In broad terms, deepfake detection can be seen as an OOD detection task in the media space $\mathcal{X}$. In contrast, semantic calibration functions exclusively in the semantic latent space $\mathcal{Z}$, a choice motivated by the fundamental limitations highlighted in Theorem 1. By design, media samples that have the same semantic information are treated equally by semantic calibration, regardless of there truthfulness (generated vs real). Our method is therefore *not* intended as a replacement for existing detection systems, but rather as a complementary framework. An illustrative example highlighting the difference is shown in Appendix A.9.

**Semantic calibration is *not* fact-checking.** Our approach mitigates deception by measuring deviations from a reference distribution of semantic content, rather than verifying factual correctness. This has several inherent limitations. On one hand, it can flag content that appears abnormally surprising, even if it is factually accurate. Conversely, it may allow content that is factually incorrect to pass undetected if its semantic profile aligns with patterns previously observed in real data. For these reasons, fact-checking should be addressed separately.

**Reliance on a trusted dataset $\mathcal{D}_r$.** Another central limitation of our method lies in its reliance on $\mathcal{D}_r$ to approximate the distribution of real content $p_r$. This assumes that the reference distribution is both representative and sufficiently comprehensive, a strong assumption in dynamic or underrepresented domains. However, this challenge can be mitigated by periodically updating $\mathcal{D}_r$ with new, vetted content, and by retraining or fine-tuning the reference model $\pi_{\theta_r}$. We refer to $p_r$ as the "real" distribution, but *this is not meant to define what is objectively real*. Instead, $p_r$ should represent a distribution that is broadly regarded as desirable, but we do not address the governance issue on how such a reference distribution is selected. We have primarily framed our analysis around misinformation introduced by deepfakes (hence the term "real"), but semantic calibration is versatile and is not tied to that setting. For instance, three of our experiments address the problem of political balancing, i.e., balancing streams of political tweets, political podcast segments, or political news articles. In all such cases, $\mathcal{D}_r$ is not meant to represent objective reality, it simply specifies the semantic distribution the system is designed to enforce. Without semantic calibration, the semantic distribution of the media stream is more vulnerable to adversarial users.

**Reliance on foundation models.**    Our method inherits the limitations of current captioning and language models, which may reduce the reliability of decisions in certain scenarios. Nonetheless, it employs off-the-shelf captioning models without fine-tuning, demonstrating robustness. Furthermore, benchmarks such as CapArena Cheng et al. (2025) show that advanced VLMs like GPT-4o can match or exceed human performance in detailed captioning. Future work should focus on enhancing these models for semantic calibration.

**Broader impact.**    The main risk of our method lies in the choice of the reference dataset $\mathcal{D}_r$, as any bias it contains will affect the filtered media stream. To mitigate this, we recommend transparent selection and public release of the datasets $\mathcal{D}_r$ and $\mathcal{D}$, the captioning model $\hat{f}$, and the two fine-tuned LLMs $\pi_\theta$ and $\pi_{\theta_r}$. If implemented responsibly, our approach could enable content moderation that is explainable, transparent, and scalable. Additionally, while our results primarily demonstrate the effectiveness of our approach for filtering media streams, it is unlikely that this would be its first application in practice. A more practical use would be to apply it as a flagging algorithm in content moderation, offering moderators (or even other moderation algorithms) a new method to assess the danger in new media samples.

## 7 Related work

**Fundamental limits of deepfake detection.**    While the limitations of deepfake detection are well-documented (Dolhansky et al., 2020; Wen et al., 2022; Hussain et al., 2021; Ikram et al., 2024), to the best of our knowledge, very few studies have attempted to establish theoretical bounds on deepfake detection performance limits in the context of online media streams. Most similar to our approach, Agarwal & Varshney (2019) frame deepfake detection as a hypothesis testing problem, deriving performance bounds based on robust statistics. Their analysis, however, is primarily focused on the context of generative adversarial networks (Goodfellow et al., 2014). Our work extends this line of reasoning by providing a theoretical bound applicable to a more general setting, independent of the specific generative model architecture, further motivating the need for alternative approaches as generative capabilities advance. To the best of our knowledge, we are the first to explicitly distinguish the semantic and non-semantic information in a media sample in the context of online media forensics.

**Content Moderation.**    Automated content moderation has evolved from transformer-based classifiers with lightweight toxicity heads (Lees et al., 2022), to multimodal architectures combining text and image encoders with cross-modal contrastive training (Yuan et al., 2023), and more recently to instruction-tuned multimodal assistants that directly answer whether content violates guidelines and generate explanations (Wu et al., 2024). In parallel, LLMs have opened new avenues for addressing problematic content through semantic understanding, being applied to automated fact-checking (Vykopal et al., 2024; Kotonya & Toni, 2020; Hu et al., 2024) and policy compliance classification (Kumar et al., 2024). These methods assess veracity against external knowledge, evaluate internal consistency, or identify hate speech, harassment, and other violations based on semantics rather than surface keywords. Complementary works study robustness through out-of-distribution detection, e.g., by adjusting likelihoods from generative models with input complexity measures to better separate in- and out-of-distribution data (Serrà et al., 2020). While effective for detecting policy violations, factual errors, or distributional anomalies, these approaches typically focus on discrete factuality ("Is this claim true?") or policy compliance ("Does this post violate rules?") and do not address the distributional aspect of deception as we define it, at least not directly. Our work aims to fill this gap.

## 8 Conclusion

This work offers a new perspective on media integrity by shifting the focus from detecting individual deepfakes to identifying broader semantic distortions. As generative models improve, low-level artifacts may disappear, but the risk of misleading content remains. We propose semantic calibration as a lightweight, interpretable method that complements existing deepfake detectors by targeting distributional shifts explicitly. Beyond deception reduction, this approach could also be used for alternative goals such as personalized filtering, balanced media streams, scalable moderation, and more. It lays the groundwork for information systems that prioritize what content communicates, rather than how it was created.

## REPRODUCIBILITY STATEMENT

All datasets used in our experiments (see Section 4) are publicly available and can be accessed through our HuggingFace repository (Appendix A.7.2), with preprocessing details provided in Appendix A.5. The prompts used for the captioning models are listed in Appendix A.6. All scripts, including those for LLM fine-tuning, captioning, and rejection sampling, are available in our GitHub repository (Appendix A.7.1), which also contains all hyperparameter configurations. Compute resources and hardware details are reported in Appendix A.8. Finally, the fine-tuned models corresponding to each experiment are released via our HuggingFace repository (Appendix A.7.2).

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

# A  APPENDIX

## A.1  SEMANTIC VS NON-SEMANTIC INFORMATION

We estimate how much of an image's information is semantic. Consider a small 256×256 RGB image compressed near the perceptual threshold ($\approx 0.3$ bits per pixel, as achieved by state-of-the-art methods Mentzer et al. (2020)), and suppose the semantic content of such an image can be expressed in 100 English words (this is a conservative estimate, given that the average caption in human-annotated datasets like MS-COCO Lin et al. (2014b) is only about 10 words). Using Shannon's estimate of roughly 1.3 bits per character and an average of 4.5 characters per word Shannon (1951), this amounts to $\approx 600$ bits of semantic information, only about

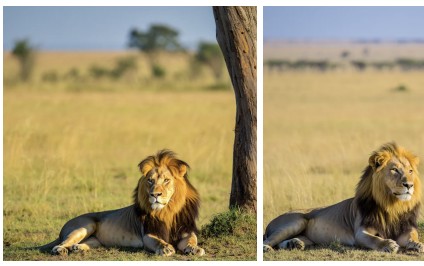

Figure 6: Two synthetic images with identical semantic information, generated by Reve.art (2025).

$600/(256^2 \times 0.3) \approx 3\%$ of the image's total perceptual information. Figure 6 shows two synthetic images that convey the same semantic information:

*A male lion lying on open savannah grassland, resting under a tree positioned to its right. Additional trees are visible in the distant background.* (24 words)

While minor differences are visible (e.g., the lion's head orientation or the exact distance of background trees), these can also be considered non-semantic, as they would likely go unnoticed if only one image were shown. Indeed, a human asked to caption a single image in Figure 6 would typically not mention such details, as they are not central to the perceived meaning of the scene. Similarly, one would most likely not specify the orientation of the lion's head while prompting a generative model.

## A.2  NOTATION

### A.2.1  VARIABLES

We define the following random variables shown in Figure 2:

- $X \in \mathcal{X}$: Observed variable representing the media of interest (e.g., text, images, video, audio, etc.).
- $Z \in \mathcal{Z}$: Latent variable representing all the semantic information present in a media $X$.
- $\hat{Z} \in \hat{\mathcal{Z}}$: Observed variable representing the textual representation of the semantic information present in a media $X$, extracted using a deterministic captioning model $\hat{f}$ such that $\hat{Z} = \hat{f}(X)$.
- $G \in \{0, 1\}$: Latent variable indicating if $X$ is generated ($G = 1$) or is real ($G = 0$).
- $F_\phi \in \{0, 1\}$: Observed variable indicating whether $X$ is flagged ($F_\phi = 1$) or not ($F_\phi = 0$). The subscript $\phi$ indicates the dependence on the decision rule $\phi(x) \triangleq \mathbb{P}[F_\phi = 1 \mid X = x]$.
- $E_\phi \in \{0, 1\}$: Latent error variable (i.e., $E = \mathbb{1}\{F_\phi \neq G\}$) with $\mathbb{1}$ the indicator variable.

### A.2.2  DISTRIBUTIONS

Let

$$p(g, z, x, \hat{g}, e) = \mathbb{P}\left[G = g, Z = z, X = x, \hat{G} = \hat{g}, E = e\right]$$

denote the joint probability distribution of the above random variables. When the context is clear, we will also use $p$ to refer to the marginal distribution over any subset of these variables (e.g., $p(x) = \mathbb{P}[X = x]$). We denote conditioning on $G = 1$ (generated media) by using $q$ in place of $p$, and conditioning on $G = 0$ (real media) by using $p_r$ instead of $p$. In addition, we use the superscript $p^\phi$ to indicate conditioning on $F_\phi = 0$ (filtered stream). More precisely:

- $p_r(x) = \mathbb{P}\left[X = x | G = 0\right]$: The distribution of the real media samples.

- $p_r(x|z) = \mathbb{P}\left[X = x | Z = z, G = 0\right]$: The distribution of real media that have semantic information $z$.

- $q(x) = \mathbb{P}\left[X = x | G = 0\right]$: The distribution of the fake media samples.

- $q(x|z) = \mathbb{P}\left[X = x | Z = z, G = 1\right]$: The distribution induced by the conditional generative model, typically a neural network (we assume that we can sample from it), for media samples that have semantic information $z$.

- $p_r(z) = \mathbb{P}\left[Z = z | G = 0\right]$: The distribution of latent semantics information found in the real media samples.

- $q(z) = \mathbb{P}\left[Z = z | G = 1\right]$: The distribution of latent semantic information specified by users when generating a media.

- $p^\phi(z) = \mathbb{P}\left[Z = z \mid F_\phi = 0\right]$: The distribution of latent semantics information found in the filtered media samples.

- $p_r^\phi(z) = \mathbb{P}\left[Z = z \mid G = 0, F_\phi = 0\right]$: The distribution of latent semantics information of the real media samples found in the filtered media stream.

- $q^\phi(z) = \mathbb{P}\left[Z = z \mid G = 1, F_\phi = 0\right]$: The distribution of latent semantics information of the generated media samples found in the filtered media stream.

In addition, we define $p_G \triangleq \mathbb{P}[G = 1]$ and $p_G^\phi \triangleq \mathbb{P}[G = 1 \mid F = 0]$ to shorten notation, and we recall the definition of the conditional entropy

$$H(Z \mid \hat{Z}) = - \sum_{\hat{z} \in \hat{\mathcal{Z}}} \mathbb{P}\left[\hat{Z} = \hat{z}\right] \sum_{z \in \mathcal{Z}} \mathbb{P}\left[Z = z \mid \hat{Z} = \hat{z}\right] \log \mathbb{P}\left[Z = z \mid \hat{Z} = \hat{z}\right].$$

### A.3 FORWARD OBJECTIVE

The objective in Eq. equation 2 is defined using the *backward* KL divergence. An alternative (but arguably harder) training objective is to minimize the *forward* KL divergence $\mathcal{L}(p_r, q)$, which encourages the generator to replicate *all* the variability present in real media. However, for our analysis, Theorem 1 also holds if one assume $\mathcal{L}(p_r, q) \leq \epsilon$ instead of $\mathcal{L}(q, p_r) \leq \epsilon$. For clarity, we only focus on the reverse KL, as it does not penalize the generator for missing low-probability modes of $p_r$, and thus tends to be easier to minimize. Indeed, training a generator that only generates convincing media samples is arguably easier than training a generator that can generate *any* type of real media.

### A.4 PROOFS

This section presents the derivations supporting the main results of the paper.

#### A.4.1 CONVEXITY OF KL DIVERGENCE

We first recall that KL divergence is convex in its first argument for discrete distributions Cover & Thomas (2006). Let $p_1$, $p_2$, and $p_3$ be distributions over a discrete space $\mathcal{Z}$, and let $\lambda \in [0, 1]$ such that $p_\lambda(z) = \lambda p_1(z) + (1 - \lambda)p_2(z)$. We have

$$
\begin{aligned}
D_{\mathrm{KL}}(p_\lambda(z) \,\|\, p_3(z)) &= \sum_{z \in \mathcal{Z}} p_\lambda(z) \log \frac{p_\lambda(z)}{p_3(z)} \\
&= \sum_{z \in \mathcal{Z}} \left[\lambda p_1(z) + (1 - \lambda)p_2(z)\right] \log \frac{\lambda p_1(z) + (1 - \lambda)p_2(z)}{p_3(z)} \\
&\leq \lambda \sum_{z \in \mathcal{Z}} p_1(z) \log \frac{p_1(z)}{p_3(z)} + (1 - \lambda) \sum_{z \in \mathcal{Z}} p_2(z) \log \frac{p_2(z)}{p_3(z)} \\
&= \lambda D_{\mathrm{KL}}(p_1(z) \,\|\, p_3(z)) + (1 - \lambda) D_{\mathrm{KL}}(p_2(z) \,\|\, p_3(z)),
\end{aligned}
$$

where the inequality follows from Jensen's inequality applied pointwise to the convex function $x \mapsto x \log(x/p_3(z))$. We can now prove the desired inequality. From the definition of $p^\phi$ in Eq. equation 4, we have

$$p^\phi(z) = p_G^\phi \, q^\phi(z) + (1 - p_G^\phi) \, p_r^\phi(z).$$

Applying the convexity of KL divergence with $\lambda = p_G^\phi$, $p_1 = q^\phi$ and $p_2 = p_r^\phi$, we obtain

$$D_{\mathrm{KL}}(p^\phi(z) \,\|\, p_r(z)) \leq p_G^\phi \, D_{\mathrm{KL}}(q^\phi(z) \,\|\, p_r(z)) + (1 - p_G^\phi) \, D_{\mathrm{KL}}(p_r^\phi(z) \,\|\, p_r(z)).$$

By definition of $\delta$ in Eq. equation 3, this yields

$$\delta(p^\phi \mid p_r) \leq p_G^\phi \delta(q^\phi \mid p_r) + (1 - p_G^\phi)\delta(p_r^\phi \mid p_r).$$

### A.4.2 DIVERGENCE CHAIN RULE

We now establish the following intermediate result that will be instrumental in proving Theorem 1:

$$D_{\mathrm{KL}}(q(x)\|p_r(x)) \leq \mathcal{L}(q, p_r) + D_{\mathrm{KL}}(q(z)\|p_r(z)). \tag{10}$$

Starting from the definition, and observing that $q(x) = q(z, x)$ and $p_r(x) = p_r(z, x)$ for $z = f(x)$ (since $f$ is deterministic, meaning each media item $x$ maps to a unique semantic representation $z = f(x)$), and letting $\mathcal{X}(z) \triangleq x \in \mathcal{X} \mid f(x) = z$, we have

$$
\begin{aligned}
D_{\mathrm{KL}}(q(x)\|p_r(x)) &= \sum_{x \in \mathcal{X}} q(x) \log \frac{q(x)}{p_r(x)} \\
&= \sum_{z \in \mathcal{Z}} \sum_{x \in \mathcal{X}(z)} q(x) \log \frac{q(z, x)}{p_r(z, x)} \\
&= \sum_{z \in \mathcal{Z}} \sum_{x \in \mathcal{X}(z)} q(z)q(x|z) \log \frac{q(z)q(x|z)}{p_r(z)p_r(x|z)} \\
&= \sum_{z \in \mathcal{Z}} \sum_{x \in \mathcal{X}(z)} q(z)q(x|z) \log \frac{q(z)}{p_r(z)} + \sum_{z \in \mathcal{Z}} \sum_{x \in \mathcal{X}(z)} q(z)q(x|z) \log \frac{q(x|z)}{p_r(x|z)} \\
&= \sum_{z \in \mathcal{Z}} q(z) \log \frac{q(z)}{p_r(z)} \left( \sum_{x \in \mathcal{X}(z)} q(x|z) \right) + \sum_{z \in \mathcal{Z}} q(z) \left( \sum_{x \in \mathcal{X}(z)} q(x|z) \log \frac{q(x|z)}{p_r(x|z)} \right) \\
&= \sum_{z \in \mathcal{Z}} q(z) \log \frac{q(z)}{p_r(z)} \left( \sum_{x \in \mathcal{X}} q(x|z) \right) + \sum_{z \in \mathcal{Z}} q(z) \left( \sum_{x \in \mathcal{X}} q(x|z) \log \frac{q(x|z)}{p_r(x|z)} \right) \\
&= \sum_{z \in \mathcal{Z}} q(z) \log \frac{q(z)}{p_r(z)} + \sum_{z \in \mathcal{Z}} q(z) D_{\mathrm{KL}}(q(x|z)\|p_r(x|z)) \\
&= D_{\mathrm{KL}}(q(z)\|p_r(z)) + \mathbb{E}_{z \sim q(z)}[D_{\mathrm{KL}}(q(x|z)\|p_r(x|z))],
\end{aligned}
$$

where we used the fact that $q(x \mid z) = p_r(x \mid z) = 0$ for $z \neq f(x)$, and adopted the standard information-theoretic convention $0 \log\left(\frac{0}{0}\right) \equiv 0$ to extend the sums from $\mathcal{X}(z)$ to $\mathcal{X}$. By definition, we have

$$\mathbb{E}_{z \sim q(z)}[D_{\mathrm{KL}}(q(x|z)\|p_r(x|z))] \leq \sup_{z \in \mathcal{Z}} D_{\mathrm{KL}}(q(x|z)\|p_r(x|z)) = \mathcal{L}(q, p_r).$$

Combining both results, we directly obtain Eq. 10.

### A.4.3 LIMIT OF DEEPFAKE DETECTION

First, we can express the probability of error of any decision rule $\phi$ as

$$\mathbb{P}\left[E_\phi = 1\right] = \mathbb{P}\left[F_\phi = 0, G = 1\right] + \mathbb{P}\left[F_\phi = 1, G = 0\right]. \tag{11}$$

Using the factorization provided in Figure 2, we can expand each term as follows:

$$\mathbb{P}\left[F_\phi = 0, G = 1\right] = \sum_{x \in \mathcal{X}} \mathbb{P}\left[F_\phi = 0, G = 1, X = x\right]$$

$$= \sum_{x \in \mathcal{X}} \mathbb{P}\left[G = 1\right]\mathbb{P}\left[X = x \mid G = 1\right]\mathbb{P}\left[F_\phi = 0, X = x\right]$$

$$= \sum_{x \in \mathcal{X}} p_G q(x)(1 - \phi(x)).$$

Additionally,

$$\mathbb{P}\left[F_\phi = 1, G = 0\right] = \sum_{x \in \mathcal{X}} \mathbb{P}\left[F_\phi = 1, G = 0, X = x\right]$$

$$= \sum_{x \in \mathcal{X}} \mathbb{P}\left[G = 0\right]\mathbb{P}\left[X = x \mid G = 0\right]\mathbb{P}\left[F_\phi = 1, X = x\right]$$

$$= \sum_{x \in \mathcal{X}} (1 - p_G)p_r(x)\phi(x).$$

Substituting both expressions in Eq. equation 11:

$$\mathbb{P}\left[E_\phi = 1\right] = \mathbb{P}\left[F_\phi = 0, G = 1\right] + \mathbb{P}\left[F_\phi = 1, G = 0\right]$$

$$= \sum_{x \in \mathcal{X}} \left[p_G q(x)(1 - \phi(x)) + (1 - p_G)p_r(x)\phi(x)\right]$$

$$\geq \sum_{x \in \mathcal{X}} \min\{p_G\, q(x), (1 - p_G)\, p_r(x)\}. \tag{12}$$

Recall that $a + b - |b - a| = 2\min\{a, b\}$ for any $a, b \geq 0$. Given any $x, z$ and setting

$$a = p_G q(x) \qquad \text{and} \qquad b = (1 - p_G)p_r(x),$$

we can write

$$\min\{p_G q(x), (1 - p_G)p_r(x)\} = \frac{1}{2}\left[p_G q(x) + (1 - p_G)p_r(x) - |(1 - p_G)p_r(x) - p_G q(x)|\right].$$

Summing over $x$, we obtain

$$\sum_{x \in \mathcal{X}} \min\{p_G\, q(x), (1 - p_G)\, p_r(x)\} = \frac{1}{2}\left[1 - \sum_{x \in \mathcal{X}} |(1 - p_G)\, p_r(x) - p_G\, q(x)|\right]. \tag{13}$$

Next, note that the total variation distance between two distributions $p(x)$ and $q(x)$ is defined as $\mathrm{TV}[p(x), q(x)] = \frac{1}{2}\sum_{x \mathcal{X}} |p(x) - q(x)|$. The key step is to relate the sum in the right-hand side of Eq. equation 13 to $\mathrm{TV}[p(x), q(x)]$. To that end, we identify two cases:

- If $p_G \leq 1 - p_G$ (i.e., $p_G \leq \frac{1}{2}$), then

$$|(1 - p_G)\, p_r(x) - p_G\, q(x)| = |p_G(p_r(x) - q(x)) + (1 - 2p_G)\, p_r(x)|$$

$$\leq |p_G(p_r(x) - q(x))| + |(1 - 2p_G)\, p_r(x)|$$

$$= p_G|p_r(x) - q(x)| + (1 - 2p_G)\, p_r(x),$$

where we have used the triangular inequality. Summing over $x$:

$$\sum_{x \in \mathcal{X}} |(1 - p_G)\, p_r(x) - p_G\, q(x)| \leq \sum_{x \in \mathcal{X}} p_G|p_r(x) - q(x)| + \sum_{x \in \mathcal{X}} (1 - 2p_G)\, p_r(x)$$

$$= 2p_G \mathrm{TV}[p_r(x), q(x)] + (1 - 2p_G).$$

Therefore:

$$\sum_{x \in \mathcal{X}} \min\{p_G \, q(x), (1 - p_G) \, p_r(x)\} = \frac{1}{2} \left[ 1 - \sum_{x \in \mathcal{X}} |(1 - p_G) \, p_r(x) - p_G \, q(x)| \right]$$

$$\geq \frac{1}{2} \left[ 1 - 2 p_G \mathbf{TV}[p_r(x), q(x)] - (1 - 2 p_G) \right]$$

$$= p_G - p_G \mathbf{TV}[p_r(x), q(x)]$$

$$= \min\{p_G, 1 - p_G\}(1 - \mathbf{TV}[p_r(x), q(x)]).$$

- If $p_G \geq 1 - p_G$ (i.e., $p_G \geq \frac{1}{2}$), then

$$|(1 - p_G) \, p_r(x) - p_G \, q(x)| = |(1 - p_G) \, [p_r(x) - q(x)] + (1 - 2 p_G) \, q(x)|$$

$$\leq |(1 - p_G) \, [p_r(x) - q(x)]| + |(1 - 2 p_G) \, q(x)|$$

$$= (1 - p_G)|p_r(x) - q(x)| + (2 p_G - 1) \, q(x).$$

Summing over $x$:

$$\sum_{x \in \mathcal{X}} |(1 - p_G) \, p_r(x) - p_G \, q(x)| = \sum_{x \in \mathcal{X}} |(1 - p_G) \, p_r(x) - p_G \, q(x)|$$

$$\leq \sum_{x \in \mathcal{X}} (1 - p_G)|p_r(x) - q(x)| + \sum_{x \in \mathcal{X}} (2 p_G - 1) \, q(x)$$

$$= 2(1 - p_G)\mathbf{TV}[p_r(x), q(x)] + (2 p_G - 1).$$

Therefore:

$$\sum_{x \in \mathcal{X}} \min\{p_G \, q(x), (1 - p_G) \, p_r(x)\} = \frac{1}{2} \left[ 1 - \sum_{x \in \mathcal{X}} |(1 - p_G) \, p_r(x) - p_G \, q(x)| \right]$$

$$\geq \frac{1}{2} \left[ 1 - 2(1 - p_G)\mathbf{TV}[p_r(x), q(x] - (2 p_G - 1) \right]$$

$$= (1 - p_G) - (1 - p_G)\mathbf{TV}[p_r(x), q(x)]$$

$$= \min\{p_G, 1 - p_G\}(1 - \mathbf{TV}[p_r(x), q(x)]).$$

Since both cases yield the same bound, we can substitute it in Eq. equation 12 to obtain

$$\inf_{\phi} \mathbb{P}\left[E_\phi = 1\right] \geq \min\{p_G, 1 - p_G\}(1 - \mathbf{TV}[p_r(x), q(x)]).$$

Using Pinsker's inequality $\mathbf{TV}[p(x), q(x)] \leq \sqrt{\frac{1}{2} D_{\mathrm{KL}}(q(x) \| p_r(x))}$, we can write

$$\inf_{\phi} \mathbb{P}\left[E_\phi = 1\right] \geq \min\{p_G, 1 - p_G\}(1 - \mathbf{TV}[p_r(x), q(x)])$$

$$\geq \min\{p_G, 1 - p_G\} \left( 1 - \sqrt{\frac{1}{2} D_{\mathrm{KL}}(q(x) \| p_r(x))} \right) \tag{14}$$

$$\geq \min\{p_G, 1 - p_G\} \left( 1 - \sqrt{\frac{\mathcal{L}(q, p_r) + D_{\mathrm{KL}}(q(z) \| p_r(z))}{2}} \right),$$

$$\geq \min\{p_G, 1 - p_G\} \left( 1 - \sqrt{\frac{\epsilon + \delta(q \mid p_r)}{2}} \right),$$

where we substituted our result from Eq. equation 10 in the second-to-last step, and then used the assumption that $\mathcal{L}(q, p_r) \leq \epsilon$ along with the definition from Eq. equation 3. Finally, we conclude the

proof as follows:

$$\sup_\phi \mathbb{P}\left[E_\phi = 0\right] = 1 - \inf_\phi \mathbb{P}\left[E_\phi = 1\right]$$

$$\leq 1 - \min\{p_G, 1 - p_G\}\left(1 - \sqrt{\frac{\epsilon + \delta(q \mid p_r)}{2}}\right)$$

$$= 1 - \min\{p_G, 1 - p_G\} + \min\{p_G, 1 - p_G\}\sqrt{\frac{\epsilon + \delta(q \mid p_r)}{2}}$$

$$\leq \max\{p_G, 1 - p_G\} + \max\{p_G, 1 - p_G\}\sqrt{\frac{\epsilon + \delta(q \mid p_r)}{2}}.$$

Note that we could have derived a similar bound with the alternative objective $\mathcal{L}'(q, p_r)$ discussed in Appendix A.3 (defined with the forward KL divergence) if we had used the alternative form of Pinsker's inequality $\mathrm{TV}[p(x), q(x)] \leq \sqrt{1/2 D_{\mathrm{KL}}(p(x)\|q(x))}$ in Eq. equation 14.

### A.4.4 REJECTION SAMPLING

We show that the semantics distribution of accepted samples $p^\phi(z)$ is exactly $p_r(z)$. Let $F \in \{0, 1\}$ be a random variable indicating if a sample with semantical information $Z$ is flagged (i.e., rejected). We have:

$$p^\phi(z) = \mathbb{P}\left[Z = z \mid F = 0\right]$$

$$= \frac{\mathbb{P}\left[Z = z, F = 0\right]}{\mathbb{P}\left[F = 0\right]}$$

$$= \frac{\mathbb{P}\left[Z = z\right]\mathbb{P}\left[F = 0 \mid Z = z\right]}{\sum_{z' \in \mathcal{Z}}\mathbb{P}\left[F = 0, Z = z'\right]}$$

$$= \frac{p(z) \cdot \frac{p_r(z)}{Mp(z)}}{\sum_{z' \in \mathcal{Z}} p(z') \cdot \frac{p_r(z')}{Mp(z')}}$$

$$= \frac{p_r(z)}{\sum_{z' \in \mathcal{Z}} p_r(z')}$$

$$= p_r(z)$$

Thus, rejection sampling produces samples exactly from the desired distribution $p_r(z)$. Note that $\mathbb{P}\left[F = 0\right] = \frac{1}{M}$, meaning that, on average, only one out of every $M$ media sample is accepted.

### A.4.5 UPPER BOUND FOR $M^*$

Assuming that $p(z) > 0$ for any $z$ and $p_G > 0$, and defining $\mathcal{Z}_+ \triangleq \{z \in \mathcal{Z} \mid p_r(z) > 0\}$, we can derive an upper bound on $M^*$ as follows:

$$M^* = \sup_{z \in \mathcal{Z}} \frac{p_r(z)}{p(z)}$$

$$= \sup_{z \in \mathcal{Z}_+} \frac{p_r(z)}{p(z)}$$

$$= \sup_{z \in \mathcal{Z}_+} \frac{p_r(z)}{p_G q(z) + (1 - p_G)p_r(z)}$$

$$= \sup_{z \in \mathcal{Z}_+} \left(\frac{1}{(1 - p_G) + p_G \cdot \frac{q(z)}{p_r(z)}}\right)$$

$$\leq \frac{1}{1 - p_G}.$$

This concludes the proof.

### A.4.6 RATIO APPROXIMATION

First, since we assume that the captioning model $\hat{f}$ is such that $H(Z \mid \hat{f}(X)) = 0$, we know that there exists a deterministic function $g : \hat{\mathcal{Z}} \to \mathcal{Z}$ such that $Z = g(\hat{f}(X))$. Assuming that the conditional generative model perfectly captures reality (i.e., $q(x' \mid z) = p_r(x' \mid z)$ for any $x', z$), we have:

$$\frac{p_r(\hat{z})}{p(\hat{z})} = \frac{\sum_{x' \in \mathcal{X}(\hat{z})} p_r(x')}{\sum_{x' \in \mathcal{X}(\hat{z})} p(x')} \qquad \mathcal{X}(\hat{z}) \triangleq \{x \mid \hat{f}(x) = \hat{z}\}$$

$$= \frac{\sum_{x' \in \mathcal{X}(\hat{z})} p_r(f(x'), x')}{\sum_{x' \in \mathcal{X}(\hat{z})} p(f(x'), x')} \qquad p_{(r)}(x') = p_{(r)}(z', x') \text{ for } z' \triangleq f(x')$$

$$= \frac{\sum_{x' \in \mathcal{X}(\hat{z})} p_r(g(\hat{f}(x')), x')}{\sum_{x' \in \mathcal{X}(\hat{z})} p(g(\hat{f}(x')), x')} \qquad f(x') = g(\hat{f}(x'))$$

$$= \frac{\sum_{x' \in \mathcal{X}(\hat{z})} p_r(g(\hat{z}), x')}{\sum_{x' \in \mathcal{X}(\hat{z})} p(g(\hat{z}), x')} \qquad \hat{f}(x') = \hat{z} \text{ for } x' \in \mathcal{X}(\hat{z})$$

$$= \frac{\sum_{x' \in \mathcal{X}(\hat{z})} p_r(z, x')}{\sum_{x' \in \mathcal{X}(\hat{z})} p(z, x')} \qquad z = g(\hat{z})$$

$$= \frac{\sum_{x' \in \mathcal{X}(\hat{z})} p_r(x' \mid z) p_r(z)}{\sum_{x' \in \mathcal{X}(\hat{z})} p(x \mid z) p(z)}$$

$$= \frac{\sum_{x' \in \mathcal{X}(\hat{z})} p_r(x' \mid z) p_r(z)}{\sum_{x' \in \mathcal{X}(\hat{z})} (p_G q(x' \mid z) + (1 - p_G) p_r(x' \mid z)) p(z)} \qquad \text{Fig. 2}$$

$$= \frac{\sum_{x' \in \mathcal{X}(\hat{z})} p_r(x' \mid z) p_r(z)}{\sum_{x' \in \mathcal{X}(\hat{z})} p_r(x' \mid z) p(z)} \qquad q(x' \mid z) = p_r(x' \mid z)$$

$$= \frac{p_r(z)}{p(z)}.$$

This concludes the proof.

### A.5 DATASETS AND LABELING DETAILS

We describe the datasets used in our experiments, including their construction, preprocessing, and the labeling schemes relevant for evaluating semantic calibration under distributional shift. Note that the numbers reported in Table 1 for $|\mathcal{D}|$ are 80% of those reported for the training set below since 20% of the training data is kept as validation (and thus excluded in $\mathcal{D}$ and $\mathcal{D}_r$). See Appendix A.7.2 for details about the exact semantic splits.

**Senator Tweets.** We use the Senator Tweets dataset KristinCoyote (2025), where each instance is created by cleaning the tweet text of any links or emojis. The labels used are *conservative* and *liberal*, depending on the senator's political affiliation. The final dataset's size is 79.2k rows for training and 19.6k for testing.

**Political Podcasts.** We ise a subset of the Political Podcasts dataset nbandhi (2025). The dataset consists of segments from podcasts with political discussions. The existing labels which we use are *conservative* and *liberal*, depending on the context. The final dataset's size is 11.4k rows for training and 1k for testing.

**Political Bias.** We use a subset of the Political Bias Corpus surajkarakulath (2025), where each instance is a news article, labeled as *left*, *center* or *right*. The final training dataset's size is 9.59k and we evaluate on 2.3k samples.

**AG News.** We use a subset of the AG News dataset Zhang et al. (2015), where each instance is created by concatenating the article's *title* and *description*. This subset is limited to the four most common categories: *World*, *Sports*, *Business*, and *Sci/Tech*. It consists of 120,000 samples in the

training set and 6,700 samples in the test set. The original four-class categorization is retained for our experiments.

**Gutenberg.** This dataset is derived from eight books sourced from the Project Gutenberg archive Project Gutenberg (2025): *Pride and Prejudice*, *Frankenstein*, *The Great Gatsby*, *The Odyssey*, *Moby Dick*, *Meditations*, *Oliver Twist*, and *War and Peace*. Each text is segmented into semantically coherent spans of 50–100 words using `nltk.tokenize`, followed by post-processing (trimming or merging) to enforce the desired range. Tokens matching non-linguistic patterns such as "--", "__", and various brackets are removed. To ensure balance, we sample uniformly across books, resulting in a dataset where each book contributes equally. The final dataset is split stochastically into 5,120 and 1,280 samples for training and testing, respectively. Phrases are labeled by their book of origin.

**COCO.** We use a subset of the Microsoft COCO dataset Lin et al. (2014a), which contains images annotated with objects from 80 predefined categories. From the original dataset, we select the first 68,800 images and generate captions using Florence Yuan et al. (2021). We further split the dataset 55,000 training samples and 13,800 test samples. Samples are assigned a binary label based on the presence of objects typically associated with physical activity: *bicycle*, *frisbee*, *snowboard*, *sports ball*, *kite*, *baseball bat*, *baseball glove*, *skateboard*, *surfboard*, *tennis racket*. Images with at least one such object are assigned a label of 1, and 0 otherwise.

**CIFAR-10.** The CIFAR-10 dataset Krizhevsky (2009) contains 60,000 images drawn from 10 distinct object types, uniformly distributed. We use the standard training and test splits of 50,000 and 10,000 images, respectively. We define the semantic shifts using the original 10 labels and caption the images using Florence Yuan et al. (2021).

**CIFAR-100.** CIFAR-100 Krizhevsky (2009) extends the structure of CIFAR-10 to 100 fine-grained object types grouped under 20 supercategories. Each fine-grained type appears with uniform frequency (1%), and coarse-level groupings appear at 5%. We use the canonical training/test split of 50,000 and 10,000 samples. To conduct distribution shifts we use coarse labels, whereas evaluation is performed on the lower-level semantics of fine labels, which makes the task more challenging.

**ImageNet.** We sample the first 127,000 images from the ILSVRC-2012 ImageNet dataset Deng et al. (2009) and generate captions using Florence Yuan et al. (2021). The resulting set is partitioned into training 101,600 training samples and 25,400 test samples. To introduce a higher-level semantic structure, we map the original 673 fine-grained labels to a custom taxonomy of 20 coarse categories (see Appendix A.7.2 for details). These coarse labels are used to induce structured semantic variation and evaluate calibration behavior under abstraction.

**Hateful Memes.** We use the full Hateful Memes dataset (Kiela et al., 2021) dataset. It comes with existing labels (hateful or non-hateful). We use Florence to caption the images and concatenate it with the provided text. The training dataset size is 6.8k and the test dataset size is 1.7k.

**Activity Net.** We use the full Activity Net dataset (Caba Heilbron et al., 2015) dataset. We map all *nodeName* labels to their *parentName* and obtain a training dataset of size 11.9k with 52 different coarse labels. The captions are taken directly from (Krishna et al., 2017).

**UrbanSound8k.** We use the full UrbanSound8k dataset Salamon et al. (2014), which consists of audio recordings from 10 acoustic event categories (e.g., *dog bark*, *children playing*, *gun shot*, etc.). As the dataset does not include an official train/test split, we partition it ourselves using an 80/20 random split. Each audio sample is captioned using Qwen-Audio Chu et al. (2023) with the prompt: *"Describe this audio."* The resulting captions are used to evaluate semantic calibration in the auditory modality.

**VGG Sound.** We construct this dataset from the original VGG Sound collection Chen et al. (2020) by mapping its 309 fine-grained acoustic categories to a custom taxonomy of 14 coarse labels (see Appendix A.7.2 for details). To avoid redundant computation, we reuse the natural language captions provided in Mei et al. (2023), specifically the model's *answer-1* to *question-1* for each sample.

### A.6 CAPTIONING MODEL PROMPTS.

- **Text Modality:** no prompt is necessary since no captioning model is used. Text samples are used directly as input.

- **Image Modality:** we pass the standard prompt: `<MORE_DETAILED_CAPTION>` to Florence Yuan et al. (2021).

- **Audio Modality:** we pass the prompt: `"Describe the sounds in this clip"` for VGGSound and `"Describe this audio"` for UrbanSound8k.

- **Video Modality:** we use the generated captions from the "Dense-Captioning Events in Videos" paper (Krishna et al., 2017) .

### A.7 REPRODUCIBILITY

#### A.7.1 CODE AND IMPLEMENTATION

All code necessary to reproduce our experiments is available at `https://github.com/BeyondDeepFakeDetection/Beyond-Deep-Fake-Detection`.

The repository includes:

- The scripts that were used for image and audio captioning, and to semantically split books into phrases.

- The script used to generate distorted distribution datasets given the general dataset.

- The LLM fine-tuning script, with all hyperparameters matching those reported in the main text.

- The JSON files with custom low- to high-level label mappings, that were used in the dataset engineering part.

- The script used to compute the per-token and total probability of a text under a fine-tuned LLM.

- A standalone implementation of our *semantic calibration* pipeline, with:
  - Rebalancing of the probability mass using $\rho$.
  - Online filtering mechanism using rejection sampling.
  - Performance metrics computation (e.g. Deception Reduction).

#### A.7.2 DATASETS AND FINE-TUNED MODELS

All datasets and fine-tuned models utilized in our experiments are publicly accessible via our Hugging Face repository:

$$\text{https://huggingface.co/BeyondDeepFakeDetection.}$$

Each dataset includes: general training set, general test set, baseline split training set, mild split training set, moderate split training set, and severe split training set. Each fine-tuned model has been trained on its corresponding training split.

### A.8 COMPUTE RESOURCES

We detail the compute requirements for each component of our pipeline to support reproducibility.

- **LLM fine-tuning (GPT-2 Small)**: Fine-tuning GPT-2 (124M parameters) on a dataset of $\sim$ 50k samples such as COCO no sports, for 5 epochs, was completed in less than 2 hours on a single NVIDIA RTX 2080 GPU.

- **Florence captioning model inference**: Caption generation using Microsoft's large Florence model was performed on an NVIDIA RTX A6000 GPU (48GB VRAM). Processing a dataset of 10k images requires approximately 2.5 hours.

- **Qwen-Audio captioning inference**: Audio captioning with Qwen-Audio was run on an NVIDIA RTX A6000. Captioning the UrbanSound8k dataset (8.6k rows) required approximately 2.5 hours.

- **LLM inference (per-token log probabilities)**: For each test dataset, we computed token-wise log probabilities using a fine-tuned GPT-2 model. Inference over the evaluation set took less than $\sim15$ minutes even on our largest datasets ($> 35k$ rows) using a single NVIDIA RTX 2080 GPU.

- **Rejection sampling process**: Our semantic calibration mechanism performs 1000 independent rejection sampling runs to compute statistics (mean, std) for distribution alignment. This process is CPU-only, completes in under 5 minutes even for the largest datasets and has negligible compute cost.

## A.9 SEMANTIC CALIBRATION VERSUS DEEPFAKE DETECTION

We illustrate the difference between semantic calibration and deepfake detection in Fig. 7.

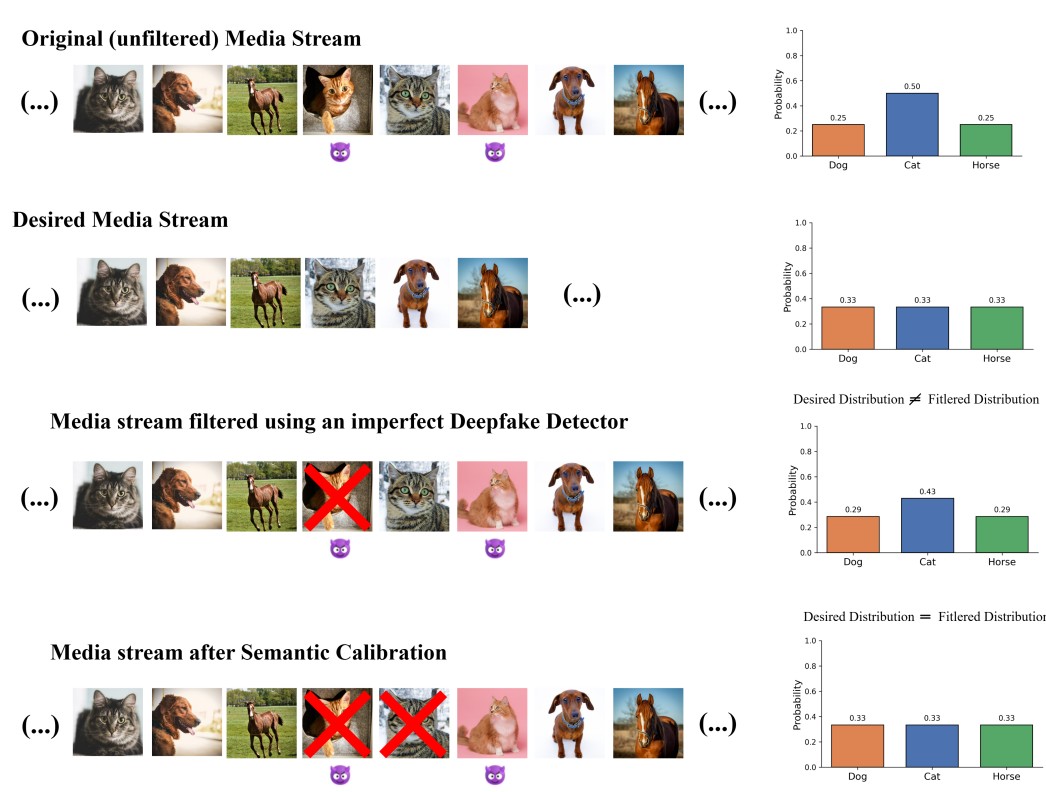

Figure 7: We illustrate an unfiltered media stream containing images of dogs, cats, and horses. The stream induces a non-uniform distribution $p$, as 2 of the 4 cat images (50%) are deepfakes (marked with the evil emoji). Suppose the deepfake detector is imperfect, as Theorem 1 suggests any detector may eventually be, and removes only 1 of the 2 fake cat images. The filtered distribution therefore remains skewed relative to the desired target. In contrast, applying our semantic calibration framework in the latent semantic space correctly adjusts the distribution $p_r$ by removing two cat instances regardless of whether they are fake or real. This example shows that our method is not a conventional deepfake detector: even when detection fails, semantic calibration rebalances manipulated media streams toward the target distribution.

## A.10 ADDITIONAL FIGURES FOR CALIBRATION PERFORMANCE

In Fig. 8 and Fig. 9, we present two additional figures that complement Fig. 4 from the main text. These provide further insight into the behavior of the distributions under different dataset shift scenarios. To further illustrate the dynamics of our method in an online setting, we provide a live animation available in the GitHub repository listed in Appendix A.7.1.

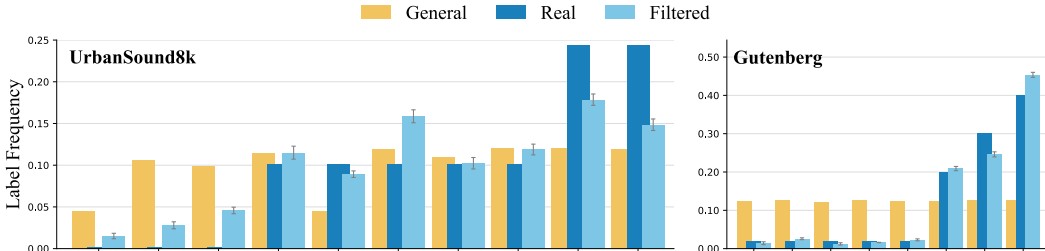

Figure 8: Comparison of the **filtered distribution** $p^\phi(z)$, the **real distribution** $p_r(z)$, and the **general distribution** $p(z)$ under a *severe shift* scenario. Results are shown for the UrbanSound8K and Gutenberg datasets. Class labels on the horizontal axis are ordered by increasing frequency in $p_r(z)$.

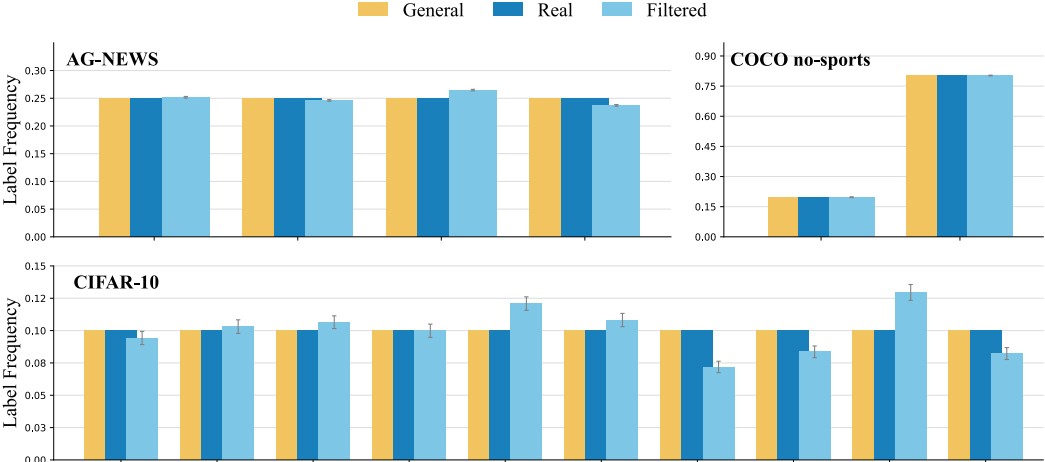

Figure 9: Visualization of the **filtered distribution** $p^\phi(z)$, the **real distribution** $p_r(z)$, and the **general distribution** $p(z)$ under a *baseline shift* scenario. Shown are results for AG-NEWS, COCO (no-sports), and CIFAR-10. Class labels along the horizontal axis are ordered by increasing frequency under $p_r(z)$. Our approach introduces minimal distortion, as expected for baseline scenarios.

## A.11 ADDITIONAL FIGURES FOR EXPLAINABILITY

To complement Fig. 1 and Fig. 5, we show additional saliency maps in Fig. 10, Fig. 12 and Fig. 14 with their corresponding rolling acceptance probabilities in Fig. 11, Fig. 13 and Fig. 15, respectively.

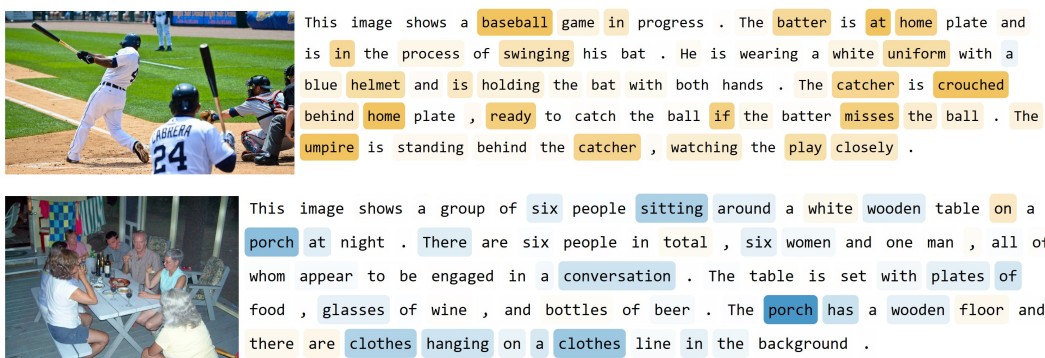

Figure 10: Saliency map showcasing the explainability of semantic calibration on two test images from the COCO dataset. Tokens highlighted in **blue favor acceptance** ($\Delta_i > 0$) while those highlighted in **orange favor rejection** ($\Delta_i < 0$). As expected, words highlighted in orange intuitively favor rejection (e.g., `baseball`, `batter`, `catcher`, `umpire`) and those in blue favor acceptance (e.g. `sitting`, `conversation`, `porch`).

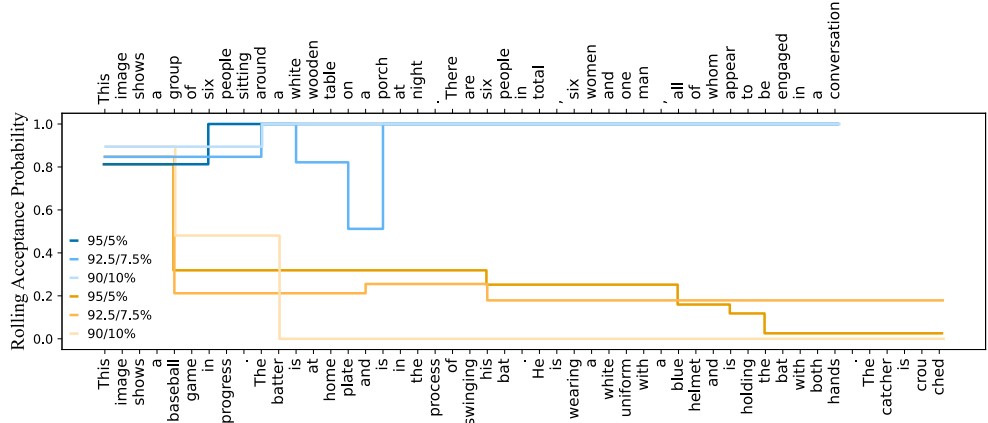

Figure 11: Rolling acceptance probabilities for the two images shown in Fig. 10

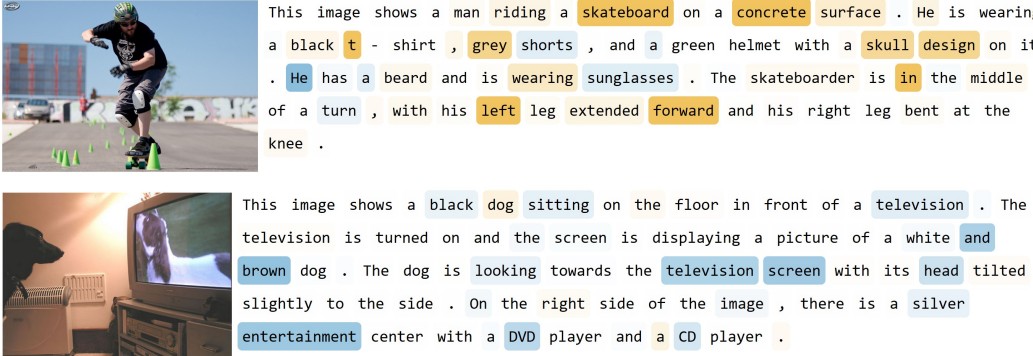

Figure 12: Saliency map showcasing the explainability of semantic calibration on two test images from the COCO dataset. Tokens highlighted in **blue favor acceptance** ($\Delta_i > 0$) while those highlighted in **orange favor rejection** ($\Delta_i < 0$). As expected, words highlighted in orange intuitively favor rejection (e.g., `skateboard`, `forward`) and those in blue favor acceptance (e.g. `television`, `screen`, `entertainment`).

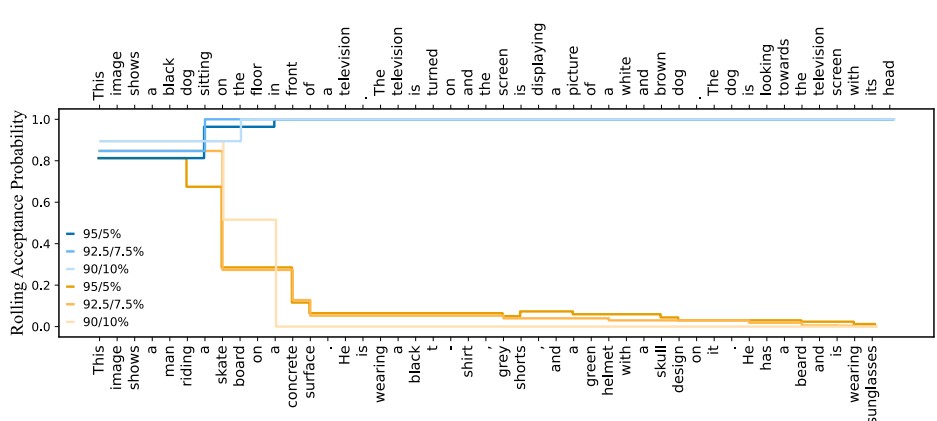

Figure 13: Rolling acceptance probabilities for the two images shown in Fig. 12

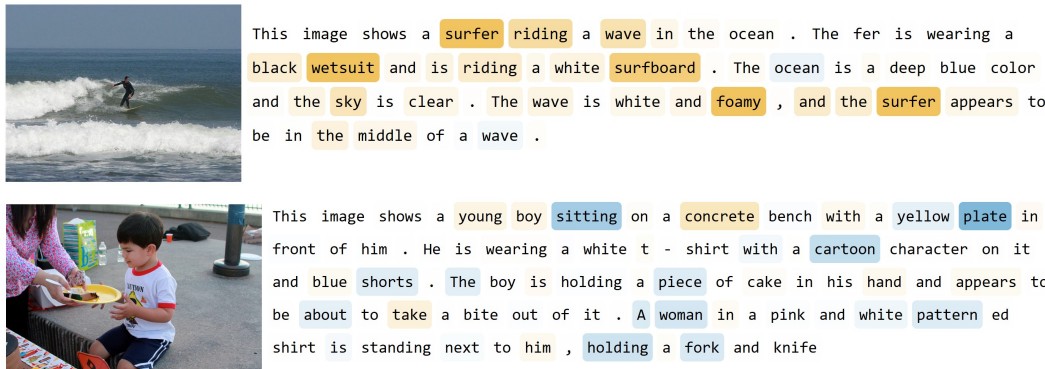

Figure 14: Saliency map showcasing the explainability of semantic calibration on two test images from the COCO dataset. Tokens highlighted in **blue favor acceptance** ($\Delta_i > 0$) while those highlighted in **orange favor rejection** ($\Delta_i < 0$). As expected, words highlighted in orange intuitively favor rejection (e.g., surfer, wetsuit, surfboard) and those in blue favor acceptance (e.g. sitting, plate, fork).

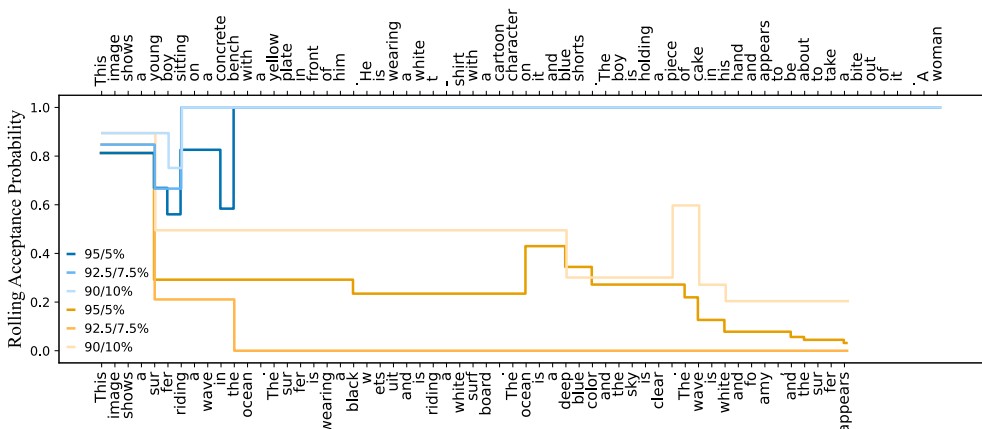

Figure 15: Rolling acceptance probabilities for the two images shown in Fig. 14

