# OpenReview forum: "Semantic Calibration in Media Streams"
_ICLR.cc/2026/Conference — ICLR 2026 Conference Desk Rejected Submission_

### Official Review · Reviewer_aoBm · 2025-10-27

**Soundness:** 2
**Presentation:** 2
**Contribution:** 2
**Rating:** 4
**Confidence:** 4

**Summary:**

This paper argues that instead of merely detecting whether media content is synthetic or not, one should directly calibrate its semantic distribution to mitigate potential semantic deception. The authors theoretically demonstrate that conventional deepfake detection methods based on non-semantic artifacts will ultimately fail as generative models approach perfection. Accordingly, they propose a Semantic Calibration framework, which employs captioning models and language models to perform rejection sampling in the semantic space, thereby enabling cross-modal content filtering.

**Strengths:**

1.The authors are the first to formally define semantic deception and introduce a distribution-level metric for it, providing a perspective that transcends the traditional binary paradigm of authenticity detection.
2.The proposed pipeline—extracting semantics via captioning models, estimating semantic distribution ratios with two language models, and performing rejection sampling—is concise, transparent, and interpretable through token-level saliency analysis.

**Weaknesses:**

1.The image experiments are conducted only on COCO, CIFAR-10, CIFAR-100, and ImageNet. It is recommended that the paper include evaluations on datasets more representative of security-related applications, such as those in the deepfake or AIGC-generated content detection domains.
2.In practical scenarios, a single image may correspond to multiple valid descriptions (e.g., “a man playing guitar” vs. “a musician performing on stage”), and a text segment may have several semantically equivalent paraphrases. Such multi-description diversity introduces representational ambiguity in the semantic space, which may cause the model to misinterpret natural linguistic variability as semantic shift or deception.
3 the method assumes captions are an (almost) lossless stand-in for semantics (aiming for 𝐻(𝑍∣𝑍^)=0) and then operates entirely in caption space; this is acknowledged but not validated rigorously, and failures of the captioner (omissions, hallucinations, bias) directly impact decisions.
4 By design the method may pass factually wrong but distribution-typical content, and reject surprising yet true items; this limits suitability for many moderation goals and shifts risk to how 𝑝𝑟 is chosen.

**Questions:**

1. How often does 𝐻(𝑍∣𝑍^)materially deviate from zero in practice? Do you have end-to-end ablations showing deception-reduction vs. caption quality (e.g., swapping Florence with a weaker/stronger captioner)?
2. If an attacker crafts captions (or ASR prompts) to mimic 𝑝𝑟(𝑧^), can they evade calibration? Any defenses beyond top-ρ filtering?
3.Your experiments mostly reweight label distributions. How does the method handle semantic recombination (e.g., rare object–context pairs) or multi-label captions?
4.Can the framework extend to video (temporal narratives) or mixed media posts where text and image semantics conflict?
5 In the “disjoint support” case you match OOD detection and report near-perfect scores on misinformation datasets; how realistic is this assumption outside curated text benchmarks?

---

> ### Author Response · Authors · 2025-11-20
>
> We thank the reviewer for their valuable feedback. In the following we address each concern.
>
> > The image experiments are conducted only on COCO, CIFAR-10, CIFAR-100, and ImageNet. It is recommended that the paper include evaluations on datasets more representative of security-related applications, such as those in the deepfake or AIGC-generated content detection domains.
>
> We emphasize that semantic calibration and synthetic media detection are disjoint tasks (see paragraph _Semantic calibration is not deepfake detection_ in Section 6). Could the reviewer clarify which experiments on AIGC-generated content they have in mind for testing semantic calibration?
>
> > In practical scenarios, a single image may correspond to multiple valid descriptions (e.g., “a man playing guitar” vs. “a musician performing on stage”), and a text segment may have several semantically equivalent paraphrases. Such multi-description diversity introduces representational ambiguity in the semantic space, which may cause the model to misinterpret natural linguistic variability as semantic shift or deception.
>
> Interestingly and somewhat counter-intuitively, we find that this is not an issue. We kindly refer the reviewer to Section 3 (Estimating $p_r(z)/p(z)$) and Appendix A.4.6 for a formal proof.
>
> > The method assumes captions are an (almost) lossless stand-in for semantics (aiming for 𝐻(𝑍∣𝑍^)=0) and then operates entirely in caption space; this is acknowledged but not validated rigorously, and failures of the captioner (omissions, hallucinations, bias) directly impact decisions.
>
> We fully agree that robust captioning is an important open problem as discussed in section 6 of our manuscript. We discuss this aspect in more detail in the general response.
>
> > By design the method may pass factually wrong but distribution-typical content, and reject surprising yet true items; this limits suitability for many moderation goals and shifts risk to how 𝑝𝑟 is chosen.
>
> Content moderation is a multi-faceted objective. In the same way that deepfake detection and fact-checking are distinct tasks, semantic caibration is distinct to both. Both deepfake detection and fact-checking work at the sample level and are incapable of calibrating distributional shifts. Consider the following simplified thought experiment: On social media, users may mostly post sightings of rare or exotic animals, while common animals like squirrels or pigeons are rarely mentioned. Every post is factually correct, but the semantic distribution is skewed toward unusual species, giving the impression that rare/endangered animals are much more common than they actually are. **Our method is the first to address these distributional shifts**. As mentioned in Section 6 of our manuscript, semantic calibration may also be used as a flagging method rather than a filtering mechanism, drastically limiting the impact of “bad” decisions.
>
>
> > How often does 𝐻(𝑍∣𝑍^)materially deviate from zero in practice?
>
> In practice, $Z$ is latent and $H(Z|\hat{Z})$ can therefore not be measured. For our experiemental setting, we test this as follows. Using 1,000 random images from *CIFAR-10*, we first generate captions for each image. We then pass each caption, along with the 10 class names to GPT-3.5, prompting it to select the class most likely represented by the caption. Finally, we compare GPT-3.5’s predicted class to the ground-truth label. This procedure achieves an accuracy of 94%, demonstrating that the captions retain the necessary semantic information for calibration in this simple setting.
>
> > Do you have end-to-end ablations showing deception-reduction vs. caption quality (e.g., swapping Florence with a weaker/stronger captioner)?
>
> We have added the requested ablation. To isolate the effect of captioning quality on semantic calibration, we run a controlled experiment on CIFAR-100 using two captioners: Florence (a lightweight VLM) and GPT-4o-mini. We test on CIFAR-100 as it is the most challenging dataset for Florence. Both captioners are evaluated under identical distribution shifts. The table below reports a head-to-head comparison across several shifts.
>
>
> | Shift      | Deception Reduction Florence | Deception Reduction GPT-4o-mini |
> |------------|----------------------------|--------------------------------|
> | Mild       | 64.81%                     | **92.82%**                        |
> | Moderate   | 80.34%                     | **83.80%**                        |
> | Severe     | 80.15%                     | **90.62%**                        |
>
>
>
> We find that stronger captioning models achieve greater deception reduction, indicating that semantic calibration will benefit from future captioning improvements.

---

> ### Author Response · Authors · 2025-11-20
>
> > If an attacker crafts captions (or ASR prompts) to mimic 𝑝𝑟(𝑧^), can they evade calibration? Any defenses beyond top-ρ filtering?
>
> In the proposed method, for non-textual media samples, captions are generated by the algorithm, not by the user/attacker. Could the reviewer clarify their question? If it concerns adverserial captioning examples, we refer the reviewer to our general response on robust captioning.
>
> > Your experiments mostly reweight label distributions. How does the method handle semantic recombination (e.g., rare object–context pairs) or multi-label captions?
>
> We would like to clarify that in our experiments, the labels are _not_ used by the method, but only for the quantitative evaluation of our semantic calibration pipeline. Captions are more expressive than labels, and therefore such settings would not pose a fundamental problem.
>
> > Can the framework extend to video (temporal narratives) or mixed media posts where text and image semantics conflict?
>
> Following the reviewer's request we apply our framework in two additional settings: **videos** and **mixed-media** where images and text jointly determine semantics.
> For the **video experiment**, we use the ActivityNet dataset [1] which consists of videos with 52 different coarse labels. We introduce semantic shifts of different strengths and obtain the following results:
>
> | Shift Level | $\delta(p\mid p_r)$ | $\delta(p^\phi\mid p_r)$ | Deception Reduction |
> |-------------|-------------------|---------------------|---------------------|
> | Baseline | 0.000 | 0.000 ± 0.000 | — |
> | Mild | 0.295 | 0.024 ± 0.001 | **91.81%** |
> | Moderate | 0.599 | 0.037 ± 0.002 | **93.84%** |
> | Severe | 0.791 | 0.051 ± 0.001 | **93.55%** |
>
> For the **mixed-media experiment**, we use the Hateful-Memes-Challenge [2]. This dataset consists of images with text, labeled as harmful or non-harmful. We use Florence to caption the iamges and concatenate the caption with the text. We apply semantic calibration under synthetic shifts that emulate different platform-level tolerance thresholds for hateful content. The results are:
>
> | Shift Level | $\delta(p\mid p_r)$ | $\delta(p^\phi\mid p_r)$ | Deception Reduction |
> |-------------|-------------------|---------------------|---------------------|
> | Baseline | 0.000 | 0.000 ± 0.000 | — |
> | Mild | 0.280 | 0.001 ± 0.000 | **99.53%** |
> | Moderate | 0.580 | 0.015 ± 0.000 | **97.40%** |
> | Severe | 0.781 | 0.043 ± 0.001 | **94.48%** |
>
> These results show that our proposed method extends to temporal captions in video and to mixed-media posts where visual and textual semantics interact. We included these experiments in Table 1 of the revised manuscript.
>
> > In the “disjoint support” case you match OOD detection and report near-perfect scores on misinformation datasets; how realistic is this assumption outside curated text benchmarks?
>
> We have added an OOD-detection evaluation on the non-curated Hateful-Memes-Challenge dataset (disjoint support with harmful or not harmful label). Below are the evaluation results, which are competitive with the challenge prize winners [3].
>
> | Accuracy | Precision | Recall | F1 Score | ROC-AUC |
> |----------|-----------|--------|----------|---------|
> | 0.8420   | 0.7294    | 0.8899 | 0.8017   | 0.8876 |
>
>
> [1] Caba Heilbron, et al. _ActivityNet: A Large-Scale Video Benchmark for Human Activity Understanding_. ICCV, 2021.
>
> [2] Kiela, D., et al. _The Hateful Memes Challenge: Detecting Hate Speech in Multimodal Memes_. arXiv:2005.04790. 2021.
>
> [3] Kiela, D., et al. _The Hateful Memes Challenge: Competition Report_. Machine Learning Research, 2021.

---

### Official Review · Reviewer_iQsp · 2025-10-28

**Soundness:** 2
**Presentation:** 2
**Contribution:** 2
**Rating:** 4
**Confidence:** 3

**Summary:**

This paper proposes semantic calibration, a novel framework that shifts deepfake and misinformation detection from artifact-based identification to semantic distribution alignment. Instead of distinguishing real versus synthetic content, the method identifies deceptive shifts in semantic information. It employs captioning models to convert multimodal inputs into text, trains two language models to estimate real versus mixed semantic distributions, and applies a likelihood-ratio–based rejection sampling rule to filter deceptive media.

**Strengths:**

1. The work introduces a distributional view of deception, formalizing it via KL divergence between semantic distributions, and rigorously establishes the limitations of traditional deepfake detection under improved generation quality. This theoretical foundation is both timely and intellectually solid.
2. The proposed semantic calibration offering transparency rarely seen in media integrity research.

**Weaknesses:**

1. Experiments are limited to classical datasets (CIFAR, COCO, AG-News, UrbanSound8K, etc.) and artificially simulated shifts, lacking tests on realistic AI-generated or manipulated content, such as GenImage [1], DeepfakeBench [2], Loki [3].
2. No quantitative or qualitative comparison is provided against advanced AI-generated content detection methods such as HAMMER (multi-modal detection) [4] or UniFD (image detection) [5], limiting the evaluation’s competitiveness.
3. The framework relies heavily on captioners or semantic encoders to extract text representations. Any bias, hallucination, or semantic drift in these upstream models directly impacts calibration reliability, but this sensitivity is not quantitatively analyzed.

[1] Genimage: A million-scale benchmark for detecting ai-generated image, NeurIPS 2023.
[2] Deepfakebench: A comprehensive benchmark of deepfake detection, NeurIPS 2024.
[3] Loki: A comprehensive synthetic data detection benchmark using large multimodal models, ICLR 2025.
[4] Detecting and grounding multi-modal media manipulation, CVPR 2023.
[5] Towards universal fake image detectors that generalize across generative models, CVPR 2023.

**Questions:**

4. Details such as model backbone choices, parameter counts, captioning prompts, and training hyperparameters are insufficiently documented.

---

> ### Author Response · Authors · 2025-11-20
>
> We thank the reviewer for their valuable feedback and address each concern in the following.
>
> > Experiments are limited to classical datasets (CIFAR, COCO, AG-News, UrbanSound8K, etc.) and artificially simulated shifts, lacking tests on realistic AI-generated or manipulated content, such as GenImage [1], DeepfakeBench [2], Loki [3].
>
> > No quantitative or qualitative comparison is provided against advanced AI-generated content detection methods such as HAMMER (multi-modal detection) [4] or UniFD (image detection) [5], limiting the evaluation’s competitiveness.
>
> We believe there might be a fundamental misunderstanding on the goal of our work. We refer the reviewer to the paragraph _Semantic calibration is not deepfake detection_ in Section 6 for more details. In particular, our manuscript states the following:
>
> > *By design, media samples that have the same semantic information are treated equally by semantic calibration, regardless of their truthfulness (generated vs real). Our method is therefore not intended as a replacement for existing detection systems, but rather as a complementary framework.*
>
> Semantic calibration and synthetic image detection are therefore not comparable. Could the reviewer clarify which exact experiments they expect on the mentioned datasets and methods?
>
> > The framework relies heavily on captioners or semantic encoders to extract text representations. Any bias, hallucination, or semantic drift in these upstream models directly impacts calibration reliability, but this sensitivity is not quantitatively analyzed.
>
> We agree that robust captioning is an important open problem. We direct the reviewer to our general response, where this point is addressed in detail.
>
> > Details such as model backbone choices, parameter counts, captioning prompts, and training hyperparameters are insufficiently documented.
>
> While some details are exluded from the main text, all implementation details are included in the appendix and all results are reproducible using the supplementary material.
>
> - Captioning prompts: Appendix A.6
> - Model backbone and parameter count: Section 4 and Appendix A.8
> - Training hyperparameters and scripts: Appendix A.7.1 and A.7.2.
>
> If the reviewer finds additional missing information, we would be happy to provide it.

---

### Official Review · Reviewer_8cL7 · 2025-10-30

**Soundness:** 3
**Presentation:** 3
**Contribution:** 3
**Rating:** 6
**Confidence:** 4

**Summary:**

The paper introduces Semantic Calibration, a theoretical and practical framework that redefines the problem of deepfake detection. Instead of distinguishing real vs. synthetic media through low-level artifacts, the authors argue that the true objective is to reduce semantic deception, which is a distributional distortions in the meaning conveyed by media streams.

The paper proves that artifact-based deepfake detection will eventually fail as generative models approach perfection. It then formalizes deception as the KL divergence between the semantic distribution of observed media and that of real data, and finally proposes a modality-agnostic mitigation strategy: converting media to text via captioning, then filtering samples using rejection sampling based on semantic likelihood ratios derived from two fine-tuned LLMs.

Extensive experiments across text, image, and audio modalities show consistent reductions in semantic deception and strong explainability via token-level saliency maps. The method is transparent and empirically effective in aligning media semantics with real-world distributions.

**Strengths:**

1. The paper makes a paradigm shift from authenticity detection to semantic distribution alignment from simple binary deepfake detection. The notion that misinformation should be treated as a semantic calibration problem is both novel and timely. This framing may become foundational for next-generation media integrity systems as artifact cues disappear

2. The authors formally derive performance bounds (Theorem 1) showing the inevitability of deepfake detection failure under improving generators. The theoretical link between deception and achievable detection accuracy is rigorous and motivates the need for semantic methods

3. The proposed rejection sampling in semantic space is mathematically clean and interpretable. Using captioning models and LLM likelihood ratios provides explainability, which is a key advantage over opaque moderation algorithms. The token level saliency analysis demonstrates further interpretability.

**Weaknesses:**

1. Because semantics are extracted via pretrained captioners (e.g., Qwen-Audio), calibration accuracy inherits their biases and failure modes. Although the authors discuss this limitation, no robustness experiments are shown under noisy or adversarial captions.

2. While experiments simulate semantic shifts via reweighted class distributions, these synthetic setups might be simplified compared to real-world misinformation, which is dynamic, adversarial, and context-dependent. Demonstrating the method on real social media streams or misinformation datasets (beyond tabular datasets) may strengthen the claim of practical viability.

3. The approach depends critically on a "trusted" dataset to model the real semantic distribution. As the authors acknowledge, this assumption is strong, where biases or incompleteness in the trusted dataset directly propagate to moderation outcomes. The paper does not provide strategies for ensuring fairness or robustness of the dataset itself.

**Questions:**

1. How would semantic calibration adapt to evolving media semantics (e.g., breaking news or new slang)? Would continual retrainingbe required to maintain the semantic distribution?

2. Given a threat model where an attacker deliberately craft content semantically close to the distribution but factually false, can calibration still filter them?

3. How feasible is deploying semantic calibration as a real-time moderation layer given captioning and LLM inference overhead?

4. Could semantic calibration be viewed as a distributional analogue of LLM alignment (e.g., minimizing semantic divergence instead of reward loss)?

---

> ### Author Response · Authors · 2025-11-20
>
> We thank the reviewer for their valuable feedback. In the following we address each question and concern seperately.
>
> > Because semantics are extracted via pretrained captioners (e.g., Qwen-Audio), calibration accuracy inherits their biases and failure modes. Although the authors discuss this limitation, no robustness experiments are shown under noisy or adversarial captions.
>
> We agree that robust captioning is an important open problem. As discussed in section 6, our proposed method inherits their limitations for modalities other than text. We direct the reviewer to our general response, where this point is addressed in detail.
>
>
> > While experiments simulate semantic shifts via reweighted class distributions, these synthetic setups might be simplified compared to real-world misinformation, which is dynamic, adversarial, and context-dependent. Demonstrating the method on real social media streams or misinformation datasets (beyond tabular datasets) may strengthen the claim of practical viability.
>
> We have added experiments on additional real-world social media datasets (see Table 1), providing evidence that our method is effective on real-world semantic shifts. We direct the reviewer to our general response, where we give more details about these additional experiements.
>
>
> > The approach depends critically on a "trusted" dataset to model the real semantic distribution. As the authors acknowledge, this assumption is strong, where biases or incompleteness in the trusted dataset directly propagate to moderation outcomes. The paper does not provide strategies for ensuring fairness or robustness of the dataset itself.
>
> We acknowledge in Section 6 that defining the trusted dataset or distribution is an open problem that we do not address. This is fundamentally a governance question, best resolved case by case and with input from all relevant stakeholders, since notions of trust, fairness and robustness vary across cultural norms and applications. Importantly, this limitation is not specific to our approach: any centralized moderation system necessarily relies on an authoritative reference (e.g., moderation guidelines, curated datasets, or moderator training material), and any bias or incompleteness in that reference will unavoidably propagate to the moderation outcomes. Moreover, semantic calibration is not restricted to centralized settings: users could also calibrate against their own selected trusted sources, producing personalized distributions rather than enforcing a universal one and therefore eliminating the issue of “fairness”.
>
> > How would semantic calibration adapt to evolving media semantics (e.g., breaking news or new slang)?
>
> The answer is somewhat counter-intuitive and we refer the reviewer to the paragraph on OOD detection in Section 6 for a formal explanation. If a media sample contains a previously unseen semantic $z$ (i.e., no sample in $\mathcal{D}$ has semantic $z$), then finetuning $\pi$ on $\mathcal{D}$ or on a subset $\mathcal{D}\_r \subseteq \mathcal{D}$ has little influence on $\pi(z)$. Consequently, the ratio $\frac{\pi_{\theta_r}(z)}{\pi_{\theta}(z)}$ should remain close to 1 (low probability to be filtered out). In essence, semantic calibration measures relative surprise with respect to a reference distribution, while OOD detection measures absolute surprise.
>
> > Would continual retraining be required to maintain the semantic distribution?
>
> In principle, continual learning is only needed if either $p_r(z)$ or $p(z)$ change over time, which depends on the application. In practice, however, any deployment of semantic calibration should include some form of continual updates to incorporate new data and maintain robustness, even in the case of stationary distributions.
>
> >Given a threat model where an attacker deliberately craft content semantically close to the distribution but factually false, can calibration still filter them?
>
> We reiterate our statement in Section 6 that semantic calibration is not fact-checking. Fact-checking relies on external knowledge sources, whereas semantic calibration operates without access to ground truth and therefore cannot verify factual accuracy. Therefore, it should **not** be assumed that semantic calibration would filter such adversarial content.
>
> > How feasible is deploying semantic calibration as a real-time moderation layer given captioning and LLM inference overhead?
>
> On 1x A100, the captioning model requires $\approx 700$ ms per sample, each LLM call $\approx 10$ ms, and rejection sampling $\approx 0.2$ ms, yielding an end-to-end latency of $\approx 710.2$ ms per instance. Leveraging further software or hardware-based optimization would likely greatly reduce the latency.

---

> ### Author Response · Authors · 2025-11-20
>
> > Could semantic calibration be viewed as a distributional analogue of LLM alignment (e.g., minimizing semantic divergence instead of reward loss)?
>
> This is an interesting question. There is indeed a conceptual parallel between semantic calibration and LLM alignment. In the same way that LLMs are generative models for text, one can view a media stream $p$ as a generative model for media content. While LLM alignment reduces the divergence between a model $\pi$ and an unknown optimal policy $\pi_r$ that maximizes an empirical reward $r$, semantic calibration reduces the divergence between an unfiltered media stream $p$ and an unknown “real” or “trusted” distribution $p_r$.

---

### Author Response · Authors · 2025-11-20

We thank the reviewers for their time spent evaluating our work and for the valuable feedback.

Reviewers generally agree that the paper presents a timely, conceptually novel shift from binary deepfake detection to semantic distribution alignment, with a rigorous theoretical foundation. They also highlight that the semantic calibration pipeline is simple, transparent, interpretable, and offers strong explainability.

The reviewers raised two main concerns. First, the method relies on captioning models, whose imperfections may limit robustness across modalities. Second, several of our experiments use controlled, synthetic semantic shifts rather than real-world media streams. In the following we address both concerns.

### Reliance on captioning models
As discussed in Sec. 6, we agree that robust captioning is an important open problem. A large body of research focuses on building strong captioning models. We refer reviewers to existing literature for more details on captioning robustness [1, 2], recent methods aimed at improving it [3], and the current state of captioners [4].

While semantic calibration inherits the limitations of captioning models for non-text modalities, it also benefits from ongoing progress in that area. We also note that the reliance on captioning models is a weakness shared by many recent automatic content moderation algorithms (e.g., [5,6]), not only semantic calibration. For instance, OpenAI's latest content moderation API uses a multimodal model `omni-moderation-latest` to classify images as harmful [6].

Finally, we emphasize that semantic calibration itself is modality-agnostic: it operates purely on text. While we use captioning models to obtain text from non-textual media, our proposed method applies equally to human-generated captions or textual content, where $Z=\hat{Z}$ and $H(Z|\hat{Z})=0$ by definition. The reliance on captioning models is therefore not intrinsic to our work.

### Additional experiments on real-world semantic shifts

Several reviewers noted that our experiments focused on simplified, artificial shifts. To address this, we added results on realistic shifts by evaluating semantic calibration on datasets of political opinions from different parties [1,2,3]. The objective is to filter a potentially skewed media stream such that the resulting stream matches a desired balanced distribution. For the binary datasets, we balance between conservative and liberal content. For the three-label dataset, we balance among left, center, and right.

We run three experiments: one on tweets from U.S. senators, one on political podcast segments, and one on bias-annotated political news (see manuscript for references). **In all cases, the objective is purely distributional: we do not attempt to fact-check or validate the content itself.** The goal is to calibrate the semantic distribution of the stream, which is the core distinction from traditional detection.

As shown below, semantic calibration consistently reduces deception by more than 80% across real semantic shifts, while introducing negligible bias on the baseline. These results have also been added to Table 1 of the manuscript.

Senator Tweets, target dist.: Conservative \(C): 50%, Liberal (L): 50%

| Case | C (Initial) | L (Initial) | C (Filtered) | L (Filtered) | Reduction |
|-|-|-|-|-|-|
| Baseline | 50% | 50% | 50% | 50% | — |
| Mild | 60% | 40% | 52% | 48% | **96.03%**|
| Moderate | 20% | 80% | 41% | 59% | **91.55%**|
| Severe | 90% | 10%| 68% | 32%| **81.99%**|

Political Podcasts, target dist.: C: 50%, L: 50%

| Case | C (Initial) | L (Initial) | C (Filtered) | L (Filtered) | Reduction |
|-|-|-|-|-|-|
| Baseline | 50% | 50% | 49% | 51% | — |
| Mild | 60% | 40% | 50% | 50% | **99.77%** |
| Moderate | 70% | 30% | 50% | 50% | **99.99%** |
| Severe | 19% | 81% | 42% | 58% | **93.79%** |

Political Bias Corpus, target dist.: Left (L): 33%, Center \(C): 33%, Right \(R): 33%

| Case | L (Initial) | C (Initial) | R (Initial) | L (Filtered) | C (Filtered) | R (Filtered) | Reduction |
|-|-|-|-|-|-|-|-|
| Baseline | 33% | 33% | 33% | 33% | 33% | 33% | — |
| Mild | 44% | 11% | 44% | 39% | 24% | 36% | **84.85%** |
| Moderate | 70% | 20% | 10% | 27% | 36% | 36% | **97.12%** |
| Severe | 75% | 10% | 15% | 49% | 22% | 29% | **84.52%** |

[1] A. Shirnin et al. Analyzing the Robustness of Vision & Language Models, IEEE/ACM Trans. Audio, Speech, Lang. Process., 2024.

[2] C. Schlarmann, M. Hein, On the Adversarial Robustness of Multi-Modal Foundation Models IEEE/CVF ICCV Workshops, 2023.

[3] S. Lee et al., Toward Robust Hyper-Detailed Image Captioning, ICML, 2025.

[4] Cheng, K. et al. Caparena: Benchmarking detailed image captioning, arXiv:2503.12329, 2025.

[5] Wu, Mengyang, et al. ICM-Assistant: Instruction-Tuning Multimodal Large Language Models for Rule-Based Explainable Image Content Moderation. AAAI, 2025.

[6] OpenAI. Upgrading the Moderation API with Our New Multimodal Moderation Model. OpenAI blog, Sept. 26, 2024.

---

### Note · Program_Chairs · 2026-01-17
**Submission Desk Rejected by Program Chairs**

The following references in this submission do not refer to real documents and/or have major errors in bibliographic information:

 Muhammad U. Ikram, Sohail J. Abuadbba, M. Ali Akber, Athar Ali, Kristen Moore, Josef Pieprzyk, and Simon S. Woo. Can Deepfake Detectors Be Trusted? A Benchmark and Robustness Analysis. IEEE Transactions on Information Forensics and Security, 2024.
Bangzhen Wen, Yiling Xu, Hongxia Wang, Han Su, Haoliang Li, Alex C. Kot, and Kwok-Yan Lam. Deepfake Detection: A Reliability-Aware Survey. arXiv preprint arXiv:2211.14001, 2022.
Yuan Mei, Mengchen Liu, Hakan Bilen, Di Tu, Ming Bai, Weidi Xie, and Andrew Zisserman. AudioSetCaps: Generating Captions for Audioset. In IEEE Winter Conference on Applications of Computer Vision (WACV), pp. 6243-6252, 2023.
Alyssa Lees, Siva Reddy Sunkara, Kiat Wee Zhang, Ala Bsharat, Marcos Zampieri, Paula Fortuna, John Pavlopoulos, Jeffrey Sorensen, Lucas Dixon, and Nithum Thain. A unified multilingual approach to toxic content classification. In Proceedings of the 60th Annual Meeting of the Association for Computational Linguistics (ACL). Association for Computational Linguistics, 2022.